# *SORBS2* is a genetic factor contributing to cardiac malformation of 4q deletion syndrome patients

Fei Liang[1,2†], Bo Wang[3†], Juan Geng[3], Guoling You[3], Jingjing Fa[3], Min Zhang[2], Hunying Sun[4], Huiwen Chen[5], Qihua Fu[3*], Xiaoqing Zhang[3*], Zhen Zhang[2*]

[1]Neonatal Intensive Care Unit, Shanghai Pediatric Congenital Heart Disease Institute and Pediatric Translational Medicine Institute, Shanghai Children's Medical Center, Shanghai Jiao Tong University School of Medicine, Shanghai, China; [2]Shanghai Pediatric Congenital Heart Disease Institute and Pediatric Translational Medicine Institute, Shanghai Children's Medical Center, Shanghai Jiao Tong University School of Medicine, Shanghai, China; [3]Shanghai Key Laboratory of Clinical Molecular Diagnostics for Pediatrics, Pediatric Translational Medicine Institute, Shanghai Children's Medical Center, Shanghai Jiao Tong University School of Medicine, Shanghai, China; [4]Key Laboratory of Pediatric Hematology and Oncology Ministry of Health and Pediatric Translational Medicine Institute, Shanghai Children's Medical Center, Shanghai Jiao Tong University School of Medicine, Shanghai, China; [5]Department of thoracic and cardiac surgery, Shanghai Children's Medical Center, Shanghai Jiao Tong University School of Medicine, Shanghai, China

*For correspondence:
zhenzhang@sjtu.edu.cn (ZZ);
qfu@shsmu.edu.cn (QF);
qingxiao18@163.com (XZ)

[†]These authors contributed equally to this work

Competing interests: The authors declare that no competing interests exist.

**Abstract** Chromosome 4q deletion is one of the most frequently detected genomic imbalance events in congenital heart disease (CHD) patients. However, a portion of CHD-associated 4q deletions without known CHD genes suggests unknown CHD genes within these intervals. Here, we have shown that knockdown of *SORBS2*, a 4q interval gene, disrupted sarcomeric integrity of cardiomyocytes and caused reduced cardiomyocyte number in human embryonic stem cell differentiation model. Molecular analyses revealed decreased expression of second heart field (SHF) marker genes and impaired NOTCH and SHH signaling in *SORBS2*-knockdown cells. Exogenous SHH rescued *SORBS2* knockdown-induced cardiomyocyte differentiation defects. *Sorbs2*[-/-] mouse mutants had atrial septal hypoplasia/aplasia or double atrial septum (DAS) derived from impaired posterior SHF with a similar expression alteration. Rare *SORBS2* variants were significantly enriched in a cohort of 300 CHD patients. Our findings indicate that *SORBS2* is a regulator of SHF development and its variants contribute to CHD pathogenesis. The presence of DAS in *Sorbs2*[-/-] hearts reveals the first molecular etiology of this rare anomaly linked to paradoxical thromboembolism.

## Introduction

Copy number variation (CNV) is a common structural variation in human genome and causes a variety of genetic syndromes. The identification of causal disease gene(s) within CNV intervals is crucial to understand the pathogenesis of the related disease. Chromosome 4q deletion syndrome is a genetic disease resulting from a chromosomal aberration that causes the missing of a portion of chromosome four long arm (*Strehle and Bantock, 2003*). Patients have a spectrum of clinical manifestations including craniofacial, cardiovascular, and gastrointestinal abnormalities, and mental and growth deficiencies (*Strehle and Bantock, 2003*). Congenital heart disease (CHD) is a common

defect seen in about half of the 4q deletion patients. A previous study narrowed the cardiovascular critical region to 4q32.2–q34.3, which contains *TLL1*, *HPGD*, and *HAND2* genes (*Xu et al., 2012*). Over-represented right-sided CHDs in 4q deletion syndrome patients suggest that *HAND2*, an essential regulator of the second heart field (SHF), is mainly responsible for the CHD phenotype (*Xu et al., 2012*; *Huang et al., 2002*). However, a part of terminal 4q deletions with CHDs that we and others have discovered does not cover *HAND2*(*Geng et al., 2014*; *Strehle et al., 2012*; *Tsai et al., 1999*). *SORBS2* within chromosomal 4q35.1 has been proposed as a candidate gene for CHD of terminal 4q deletion syndrome based on an unusual small interstitial deletion (*Strehle et al., 2012*). However, there has been no further evidence to substantiate it ever since. Here, we have presented evidence from in vitro cardiogenesis, animal model, and mutation analyses to demonstrate that *SORBS2* is a genetic factor regulating cardiac development and contributing to cardiac malformation of the CHD population.

## Results

### *SORBS2* is required for cardiomyocyte differentiation and the integrity of sarcomeric structure

To recapitulate *SORBS2* haploinsufficiency of 4q deletion, we knocked down *SORBS2* in human embryonic stem cell lines (H1-hESC). We used two different short hairpin RNAs (shRNAs) to knock down *SORBS2*, and similar knockdown efficiencies (~40% of wild-type expression level) were achieved (*Figure 1—figure supplement 1A*). *SORBS2* knockdown did not affect clone morphology, pluripotency marker expression, and apoptosis of human embryonic stem cells (hESCs) (*Figure 1—figure supplement 1B–1F*). After in vitro cardiac differentiation (*Burridge et al., 2014*; *Figure 1—figure supplement 2A*), spontaneous beating started to appear at differentiation day 8 (D8) in both control and *SORBS2*-knockdown cells, but differentiated *SORBS2*-knockdown cardiomyocytes contracted much weaker (*Videos 1–3*). Since *SORBS2*-knockdown embryonic stem cells from different shRNAs had similar phenotypes (*Figure 1—figure supplement 2B*, *Videos 1–3*), we only used *SORBS2-shRNA1* for further analyses.

The cardiomyocyte differentiation efficiency (the proportion of cTnT$^+$ cells) was significantly decreased in *SORBS2*-knockdown group at D15 (*Figure 1A–B*). Since *SORBS2* is a structural component of sarcomeric Z-line and cardiomyopathy gene (*Ding et al., 2020*; *Li et al., 2020*; *Sanger et al., 2010*), we examined the myofibril structure of differentiated cardiomyocytes. Most cardiomyocytes in *SORBS2*-knockdown group presented a round or oval shape instead of polygonal or spindle-like outlines in control group, and a close lookup indicated that sarcomeric structure in cells with abnormal shapes was disrupted (*Figure 1C*). The percentage of cardiomyocytes with well-organized sarcomeres and a normal shape was much lower in *SORBS2*-knockdown group (*Figure 1D*). Disrupted sarcomeric structures in *SORBS2*-knockdown cardiomyocytes were also present in transmission electron microscopy analysis

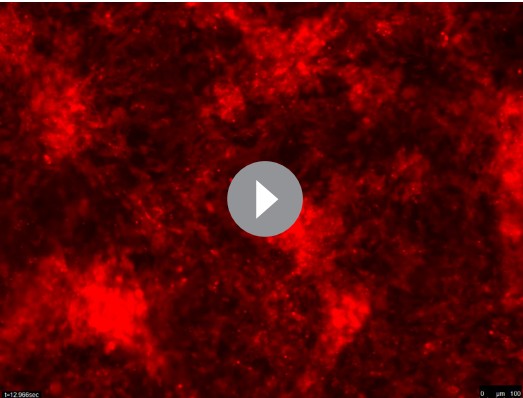

**Video 1.** Beating D20 control cardiomyocytes.
https://elifesciences.org/articles/67481#video1

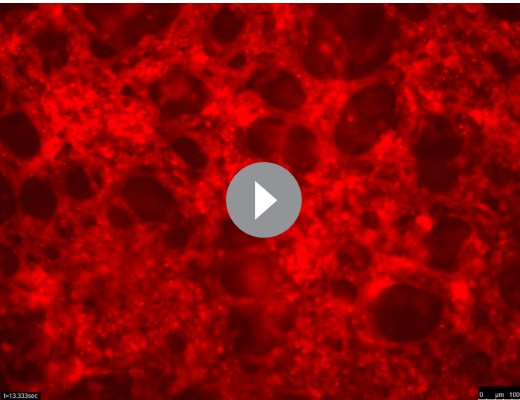

**Video 2.** Beating D20 *shRNA-SORBS2-1* cardiomyocytes.
https://elifesciences.org/articles/67481#video2

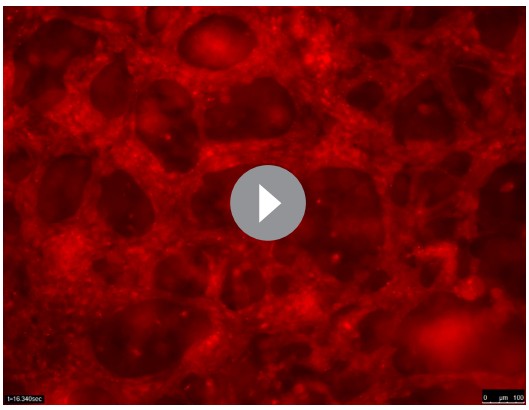

**Video 3.** Beating D20 *shRNA-SORBS2-2* cardiomyocytes.
https://elifesciences.org/articles/67481#video3

(*Figure 1—figure supplement 3A*). The expression of sarcomeric genes *TNNT2*, *MYL7*, *MYH6*, and *MYH7* was significantly decreased (*Figure 1—figure supplement 3B–3E*).

The weakened beating force of *SORBS2*-knockdown cardiomyocytes might be derived from abnormal electrophysiology. To this end, we examined the electrical activities of dissociated D30 cardiomyocytes by patch clamping. The dominant type of cardiomyocytes is ventricular-like in both control and *SORBS2*-knockdown groups (*Figure 1—figure supplement 3F*). Statistical analyses on action potential parameters of ventricular-like cells, including average action potential (AP) duration at 90% repolarization, average AP frequency, peak amplitude, and resting potential, showed no difference between two groups (*Figure 1—figure supplement 3G–3J*).

## *SORBS2* knockdown decreased the expression of SHF marker genes

Having shown the reduced efficiency of cardiomyocyte differentiation in *SORBS2*-knockdown group, we hypothesized that *SORBS2* had an early role in cardiomyocyte differentiation. Expression dynamics showed that *SORBS2* was up-regulated at cardiac progenitor stage D5 after a transient absence at mesodermal cell stage (D2-D3) (*Figure 1E*). Consistently, *SORBS2*-knockdown group disrupted the expression of cardiac progenitor markers, whereas mesodermal markers remained unchanged (*Figure 1F–H*, *Figure 1—figure supplement 4A–4E*). There are two sets of molecularly distinct cardiac progenitors during mammalian heart development, referred to as the first and second heart fields (FHF and SHF), which contribute to distinct anatomical structures of the heart (*Srivastava, 2006*). Interestingly, we found significantly increased expression of FHF markers (*TBX5*, *HCN4*, *HAND*1) (*Figure 1—figure supplement 4C–4E*) while significantly decreased expression of SHF markers (*TBX1*, *ISL1*, *MEF2C*) in *SORBS2*-knockdown cells (*Figure 1F–H*). SHF gives rise to cardiac outflow, right ventricle, and inflow (*Kelly, 2012*). Any defect in these embryonic structures leads to CHDs commonly seen in 4q deletion syndrome.

## *SORBS2* knockdown decreased NOTCH and SHH signaling

To understand how *SORBS2* regulates SHF progenitor commitment, we collected D5 cells for RNA-seq. Using a stringent threshold (padj <0.05, |log$_2$(fold change)|>1), we selected out 160 down-regulated and 104 up-regulated genes for gene ontology (GO) analysis (*Figure 1I*, *Supplementary file 1–2*). Results showed that the up-regulated genes were enriched in biological processes like cell adhesion and so on (*Figure 1J*), which might be a compensatory reaction to reduced SORBS2 as a cytoskeleton component. The down-regulated genes were enriched in biological processes like heart development and so on (*Figure 1K*), suggesting that *SORBS2* positively regulates cardiac development. Particularly, we noted the NOTCH signaling pathway in the down-regulated list (*Figure 1K–L*). We verified the expression of NOTCH signaling target genes *HEY1*, *HEYL*, and *NRARP* by qPCR (*Figure 1—figure supplement 4F*). In contrast, we did not see differential expression for *NOTCH1* in RNA-seq (*Figure 1L*), suggesting that the regulation of SORBS2 on NOTCH signaling might be through modulating protein level. SORBS2 can interact with the non-receptor tyrosine kinase c-ABL as SH3 domain-containing adaptor (*Kioka et al., 2002*). The binding of SORBS2 to c-ABL triggers the recruitment of ubiquitin ligase CBL and leads to the ubiquitination of c-ABL (*Soubeyran et al., 2003*). Indeed, we noted that c-ABL protein level was significantly elevated in *SORBS2*-knockdown cells (*Figure 1M*). c-Abl can promote Notch endocytosis to modulate Notch signaling (*Xiong et al., 2013*). Consistently, NOTCH1 protein level decreased significantly in *SORBS2*-knockdown cells (*Figure 1N*). Notch signaling is a well-known molecular mechanism enhancing cellular response to Shh (*Stasiulewicz et al., 2015*). We noted that the expression of *SHH* and SHH signaling targets,

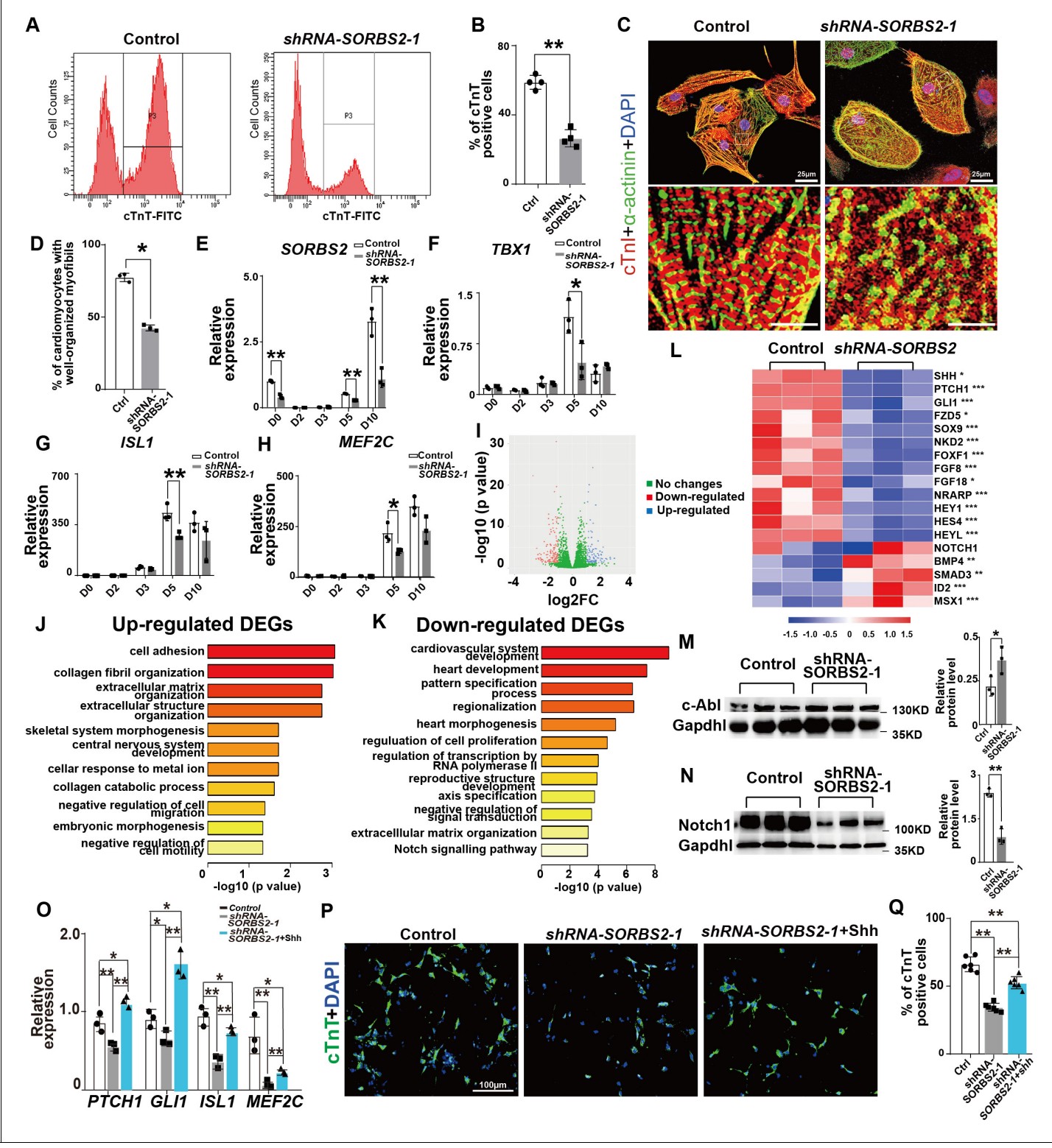

**Figure 1.** *SORBS2* has a dual role in cardiogenesis. (**A**) Flow cytometry analysis of cardiomyocytes at differentiation day 15 (D15). P3 indicates cTnT$^+$ population. (**B**) Quantification of cTnT$^+$ cells (n = 3). **p<0.01; two-tailed Student's *t* test. (**C**) Immunostaining of D30 cells with anti-cardiac troponin I (cTnI, red) and anti-α-actinin (green) antibodies. Boxed areas are magnified in the lower panels. (**D**) Quantification of cardiomyocytes with well-organized sarcomeres (control: n = 211, *SORBS2*-knockdown: n = 197). **p<0.01; two-tailed Student's *t* test. (**E**) qPCR quantification of *SORBS2* expression dynamics (n = 3 for each time point). **p<0.01; two-tailed Student's *t* test. (**F–H**) qPCR quantification of second heart field (SHF) progenitor marker expression at different time points (n = 3 for each time point). *p<0.05, **p<0.01; two-tailed Student's *t* test. (**I**) Volcano plot illustrates the

*Figure 1 continued on next page*

*Figure 1 continued*

differential gene expression from D5 RNA-seq data. Pink, down-regulated genes. Blue, up-regulated genes. (|log2(fold change)|>1 and padj <0.05). FC, fold change. (J, K) Gene ontology (GO) analysis of differentially expressed genes. Up-regulated pathways (J). Down-regulated pathways (K). DEGs, differentially expressed genes. (L) Heatmap illustrating gene expression changes of critical signaling pathways. Color tints correspond to expression levels. *padj <0.05. **padj <0.01. ***padj <0.001. (M) Western blot quantification of c-ABL expression on D5 cell lyses (n = 3). *p<0.05; two-tailed Student's *t* test. (N) Western blot quantification of NOTCH1 expression on D5 cell lyses (n = 3). **p<0.01; two-tailed Student's *t* test. (O) qPCR quantification analyses of SHH signaling target genes and SHF marker expression at D5 (n = 3 for each group). *p<0.05, **p<0.01; two-tailed Student's *t* test. (P) Representative images of immunofluorescent staining of D15 cells with anti-cardiac troponin T(cTnT, green) antibody. (Q) Quantification of cTnT⁺ cells (n = 6). **p<0.01; two-tailed Student's *t* test.

The online version of this article includes the following figure supplement(s) for figure 1:

**Figure supplement 1.** Characterization of *shRNA-SORBS2* hESCs.
**Figure supplement 2.** In vitro cardiogenesis from hESCs.
**Figure supplement 3.** Cardiomyocyte defects in *SORBS2*-knockdown cells.
**Figure supplement 4.** Molecular profiling of *SORBS2*-knockdown hESC-derived mesoderm and cardiac progenitors.

*PTCH1* and *GLI1,* was also reduced in *SORBS2*-knockdown cells (*Figure 1—figure supplement 4G*). We applied recombinant SHH protein to check whether it can rescue defects caused by *SORBS2* knockdown. As expected, exogenous SHH activated *PTCH1* and *GLI1* expression (*Figure 1O*). It also up-regulated the expression of SHF markers *ISL1* and *MEF2C* in D5 *SORBS2-shRNA1* cells (*Figure 1O*) and rescued cardiomyocyte differentiation efficiency with more cells presenting a polygonal or spindle-like shape (*Figure 1P and Q*).

## *Sorbs2⁻ᐟ⁻* mice have atrial septal defect and defective dorsal mesenchyme protrusion

Since the entire *SORBS2* gene is absent in terminal 4q deletion, we used *Sorbs2* knockout mice to examine its role in cardiac development. In a previous report, about 40–60% *Sorbs2⁻ᐟ⁻* mice died within 1 week after birth (*Zhang et al., 2016*), indicating a possible structural heart defect(s). To this end, we collected 137 embryos at E18.5. The ratio of genotype distribution among embryos was consistent with Mendel's law (*Supplementary file 3*), suggesting no embryo loss in early development stage. We dissected 30 *Sorbs2⁻ᐟ⁻* embryos, and none of them showed conotruncal defect or ventricular septal defect (*Figure 2A–B*). However, we found that about 40% (12/30) *Sorbs2⁻ᐟ⁻* hearts had atrial septal defect (ASD) with 10 being the absence/hypoplaisa of primary septum and two being double atrial septum (DAS) (*Figure 2C*, *Supplementary file 3*). The penetrance of ASD is similar to the ratio of reported postnatal lethality, indicating that ASD might contribute to early postnatal death.

A major part of atrial septum is derived from dorsal mesenchyme protrusion (DMP) originated from posterior SHF (*Kelly, 2012*). Our previous data indicates that *SORBS2* knockdown impaired the in vitro differentiation of SHF progenitors. We also noted DMP malformation in 5 out of 15 E10.5 *Sorbs2⁻ᐟ⁻* embryos. The majority of them had DMP aplasia/hypoplasia (n = 4), whereas one of them had duplicated DMP (*Figure 2D*). The dichotomy of DMP morphology is consistent with two opposite ASD phenotypes seen in E18.5 embryos. Overall, the in vivo phenotype of *Sorbs2⁻ᐟ⁻* mice further supports that *SORBS2* haploinsufficiency in 4q deletion contributes to CHD pathogenesis through affecting SHF development.

## *SORBS2* deficiency-induced molecular changes are highly conserved in mice

To examine *Sorbs2* expression pattern in early mouse embryos, we pooled publicly available single-cell transcriptomic profiles from E9.25 to 10.5 mouse embryonic hearts (*de Soysa et al., 2019*; *Hill et al., 2019*). We identified nine subgroups as cardiac progenitors or cardiomyocytes and noted that *Sorbs2* is highly expressed in cardiomyocytes and in a subgroup of cardiac progenitors that also express *Isl1* and *Tbx1* (*Figure 3—figure supplement 1*).

We used qPCR to validate molecular findings of hESCs in E10.5 mouse embryos. Since the penetrance of cardiac phenotype is about 40%, we increased the number of *Sorbs2⁻ᐟ⁻* embryos according to this ratio. Consistent with hESC differentiation results, we detected significantly down-regulated expression in three out of four Notch and Shh signaling target genes (*Hey1, Heyl*, and *Ptch1*)

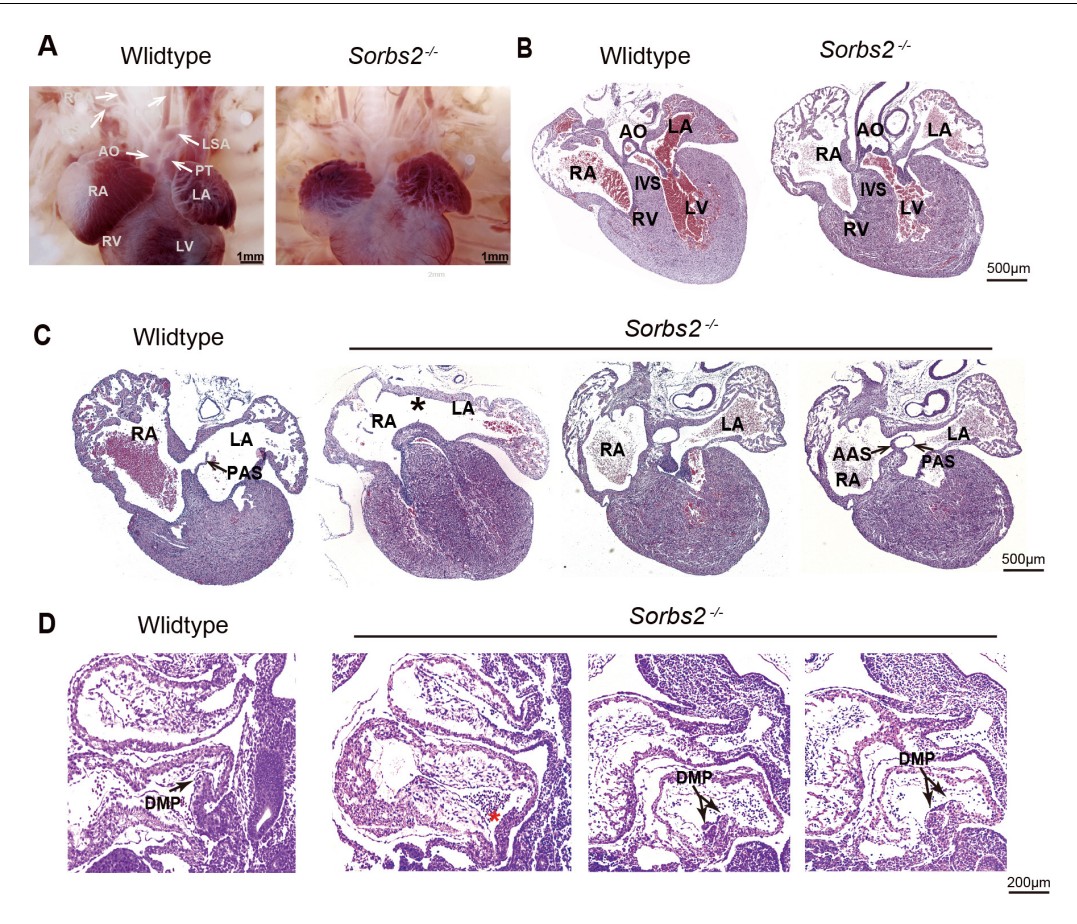

**Figure 2.** Cardiac phenotype of *Sorbs2⁻ᐟ⁻* mice. (**A**) Gross view of embryos at E18.5. (**B**) Hematoxylin and eosin (HE)-stained paraffin sections of E18.5 hearts in conotruncal area. (**C**) HE-stained paraffin sections of E18.5 heart in atrial septum area. Asterisk indicates the absence of PAS. Two sections in the right are from the same heart with double atrial septum. The rightmost section is dorsal to the other. (**D**) HE-stained paraffin sections of E10.5 embryos. Arrow indicates DMP in the atria. Red asterisk indicates hypoplastic DMP in *Sorbs2⁻ᐟ⁻* embryos. Double-headed arrow indicates a duplicated DMP in an *Sorbs2⁻ᐟ⁻* embryo. Two sections in the right are from the same embryo with duplicated DMP. The rightmost section is lateral to the other. AO, aorta. PT, pulmonary trunk. LSA, left subclavian artery. RSA, right subclavian artery. LCA, left common carotid artery. RCA, right common carotid artery. LA, left atrium. RA, right atrium. LV, left ventricle. RV, right ventricle. PAS, primary atrial septum. AAS, accessory atrial septum. IVS, interventricular septum. DMP, dorsal mesenchymal protrusion.

(*Figure 3A*). Multivariate PERMANOVA (permutational multivariate analysis of variance) revealed a significant combined difference in Notch and Shh signaling target expression between wild-type and *Sorbs2⁻ᐟ⁻* embryos (p<0.009).

To have a view of transcriptomic changes in *Sorbs2* mutants, we performed RNA-seq on E10.5 wild-type *Sorbs2⁺ᐟ⁻* and *Sorbs2⁻ᐟ⁻* embryos. Principal component analysis (PCA) indicates that wild-type embryos are clustered together, whereas *Sorbs2⁻ᐟ⁻* embryos are more scattered (±), which is consistent with diverged cardiac phenotypes of *Sorbs2⁻ᐟ⁻* embryos. Interestingly, the majority of *Sorbs2⁺ᐟ⁻* samples are juxtaposed more closely to *Sorbs2⁻ᐟ⁻* embryos in PCA plot, indicating a molecular phenotype in heterozygous mutants. We selected genes significantly down-regulated in heterozygous mutants (log2(fold change)>0.25, p<0.05) to perform GO analysis and found that these genes are enriched in pathways involved in muscle development and embryonic morphogenesis (*Figure 3C*, *Supplementary file 4*). Using the same threshold, we selected genes significantly down-regulated in homozygous mutants to perform GO analysis. These genes are enriched in pathways regulating heart development, myofibril assembly, and cardiac septum development (*Figure 3D*, *Supplementary file 5*). Particularly, the Notch signaling pathway was also in the enrichment list. Looking closer, we noted that genes involved in cardiac development, myofibril assembly, and contraction force were down-regulated in nearly all the homozygous mutants (*Figure 3E*). The decrease

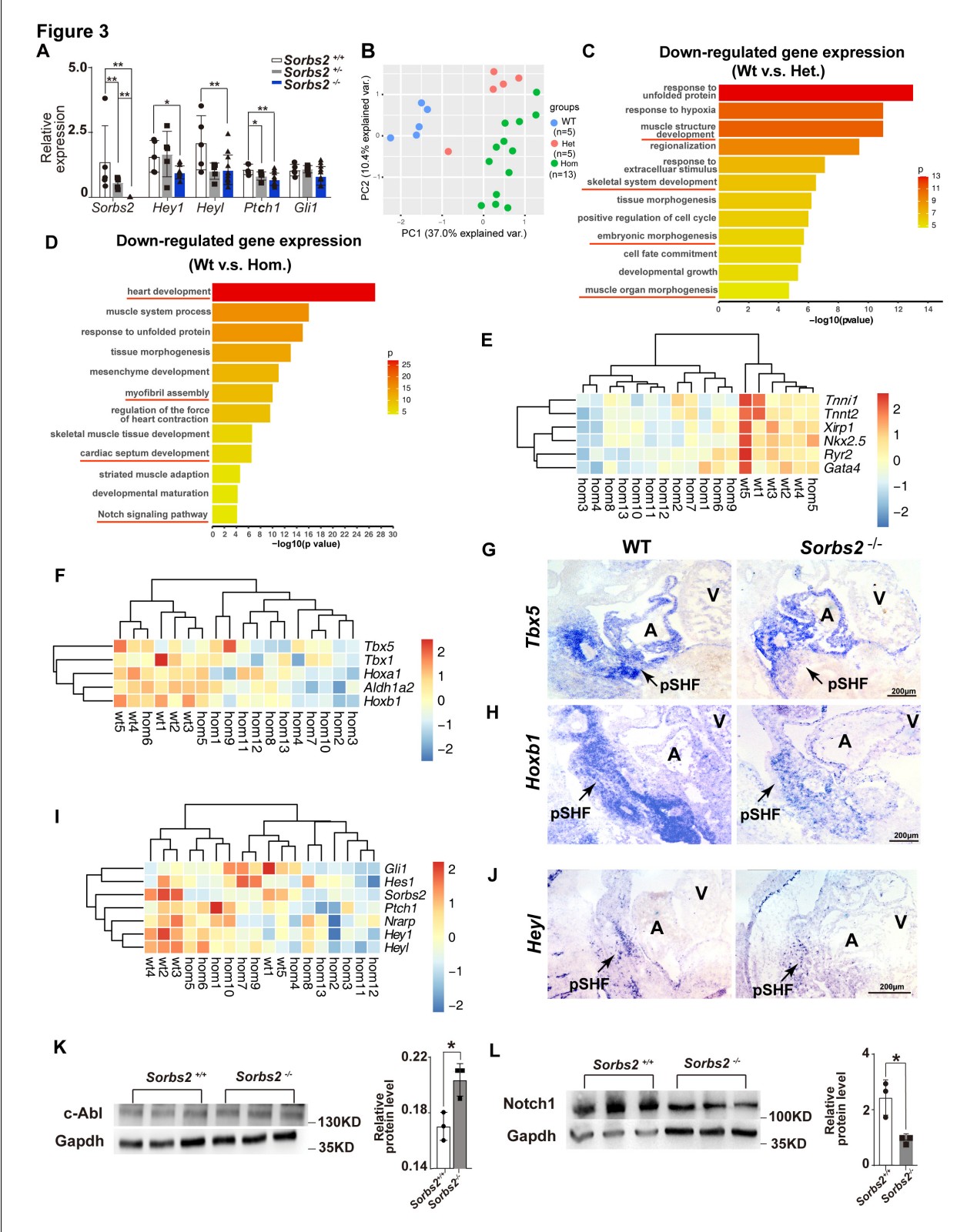

**Figure 3.** Molecular changes in *Sorbs2* mutants. (**A**) qPCR quantification of *Sorbs2*, *Hey1*, *Heyl*, *Ptch1*, and *Gli1* expression (n = 5 for wild-type and heterozygous groups, n = 13 for homozygous group). \*\*p<0.01, \*p<0.05; two-tailed Student's *t* test. (**B**) Principal component analysis (PCA) plot of RNA-seq data shows sample clustering according to genotypes. (**C**) Gene ontology (GO) analysis of genes down-regulated in heterozygous mutants (Het). (**D**) GO analysis of genes down-regulated in homozygous mutants (Hom). (**E**) Hierarchical heatmap of cardiac genes. (**F**) Hierarchical heatmap of

*Figure 3 continued on next page*

*Figure 3 continued*

posterior second heart field (SHF) markers. (**G**) RNA in situ hybridization of *Tbx5* on E10.5 embryos. (**H**) RNA in situ hybridization of *Hoxb1* on E10.5 embryos. (**I**) Hierarchical heatmap of Notch and Shh signaling genes. (**J**) RNA in situ hybridization of *Heyl* probe on E10.5 embryos. (**K**) Western blot quantification of c-Abl expression in E10 embryos. *p<0.05; two-tailed Student's *t* test. (**L**) Western blot quantification of Notch1 expression in E10.5 embryos. *p<0.05; two-tailed Student's *t* test. pSHF, posterior second heart field.

The online version of this article includes the following figure supplement(s) for figure 3:

**Figure supplement 1.** scRNA-seq analysis of *Sorbs* expression pattern in embryonic hearts.

of posterior SHF marker genes was less general but still clearly down-regulated in the majority of *Sorbs2⁻/⁻* embryos (*Figure 3F*). Decreased *Tbx5* (n = 3) and *Hoxb1* (n = 3 out of 4) expression in posterior SHF of *Sorbs2⁻/⁻* embryos was validated by RNA in situ hybridization (*Figure 3G and H*). A portion of embryos had obvious down-regulation of Notch and Shh signaling targets (*Figure 3I*), which is consistent with a low penetrance of ASD in homozygous mutants. RNA in situ hybridization confirmed decreased *Heyl* expression in posterior SHF of *Sorbs2⁻/⁻* embryos (n = 1; *Figure 3J*). Next, we verified the upstream molecular changes of Notch signaling found in hESC differentiation model. Indeed, we noted that Notch1 expression level significantly decreased whereas c-ABL significantly increased in *Sorbs2⁻/⁻* embryos (*Figure 3K–L*).

## Rare *SORBS2* variants are significantly enriched in CHD patients

Rare genetic variants play a significant role in CHD occurrence (*Blue et al., 2017*), hence we used a rare variant association to help identify genes within CNVs responsible for CHDs. Besides 23 candidate CNV genes containing *SORBS2*, the targeted panel also includes 81 known CHD genes from literature. Targeted sequencing was performed on 300 complex CHD cases (two cases removed due to low-quality data). 220 Han Chinese descents from the 1000 genome project were used as controls. The ethnicity of these two groups was matched by PCA (*Figure 4—figure supplement 1*). A total of 1560 exonic variants from CHD and control groups passed quality control and were included for further analyses (*Figure 4A*). Variant distribution in the breakdown categories of CHD and control groups is shown in *Supplementary file 6*.

Of the 847 nonsynonymous variants, 43.57% (n = 369) variants had a minor allele frequency (MAF) below 1% across ExAC database and were adjudicated as 'damaging' by at least two algorithms (PolyPhen2, SIFT, or MutationTaster). We applied gene-based statistic tests to evaluate the cumulative effects of rare damaging variants (MAF<1%) on CHDs. Genes with at least two rare damaging variants (n = 57) were included for analysis (*Supplementary file 7*). 4 out of 57 genes (*SORBS2, KMT2D, EVC2*, and *SH3PXD2B*) had a p-value lower than 0.05 (one-tailed Fisher's exact) and two of them (*SORBS2* and *KMT2D*) had a statistically significant mutation burden after the correction for multiple testing (q<0.20) (*Figure 4B*, *Supplementary file 7*). *KMT2D* is a well-known CHD gene (*Ang et al., 2016*; *Jin et al., 2017*). Our data indicate that rare *SORBS2* variants have similar levels of enrichment in CHDs as the known CHD genes. The distribution of rare *SORBS2* damaging variants in CHD patients spread throughout the gene (*Figure 4C*, *Supplementary file 8*). Although we didn't detect *SORBS2* nonsense variants in CHD patients, missense mutations identified in our cohort caused protein aggregation in cells (*Figure 4D*), suggesting an abnormal function of these variant proteins. A high prevalence (85%, 17/20) of ASD, the defect seen in *Sorbs2⁻/⁻* hearts, was observed in patients carrying *SORBS2* variants (*Supplementary file 9*). In our CHD cohort, we noted a significant enrichment of *SORBS2* rare damaging variants in patients with ASD (17 out of 183 ASD patients versus 3 out of 117 non-ASD patients, p=0.0306, Fisher's exact test). These data further support that *SORBS2* contributes to CHD pathogenesis.

## Discussion

The common CHDs in 4q deletion syndrome include ASD, ventricular septal defect (VSD), pulmonary stenosis/atresia, and tetralogy of Fallot and so on *Strehle and Bantock, 2003*; *Lin et al., 1988*. The affected structures are atrial septum and cardiac outflow tract, which are all derived from

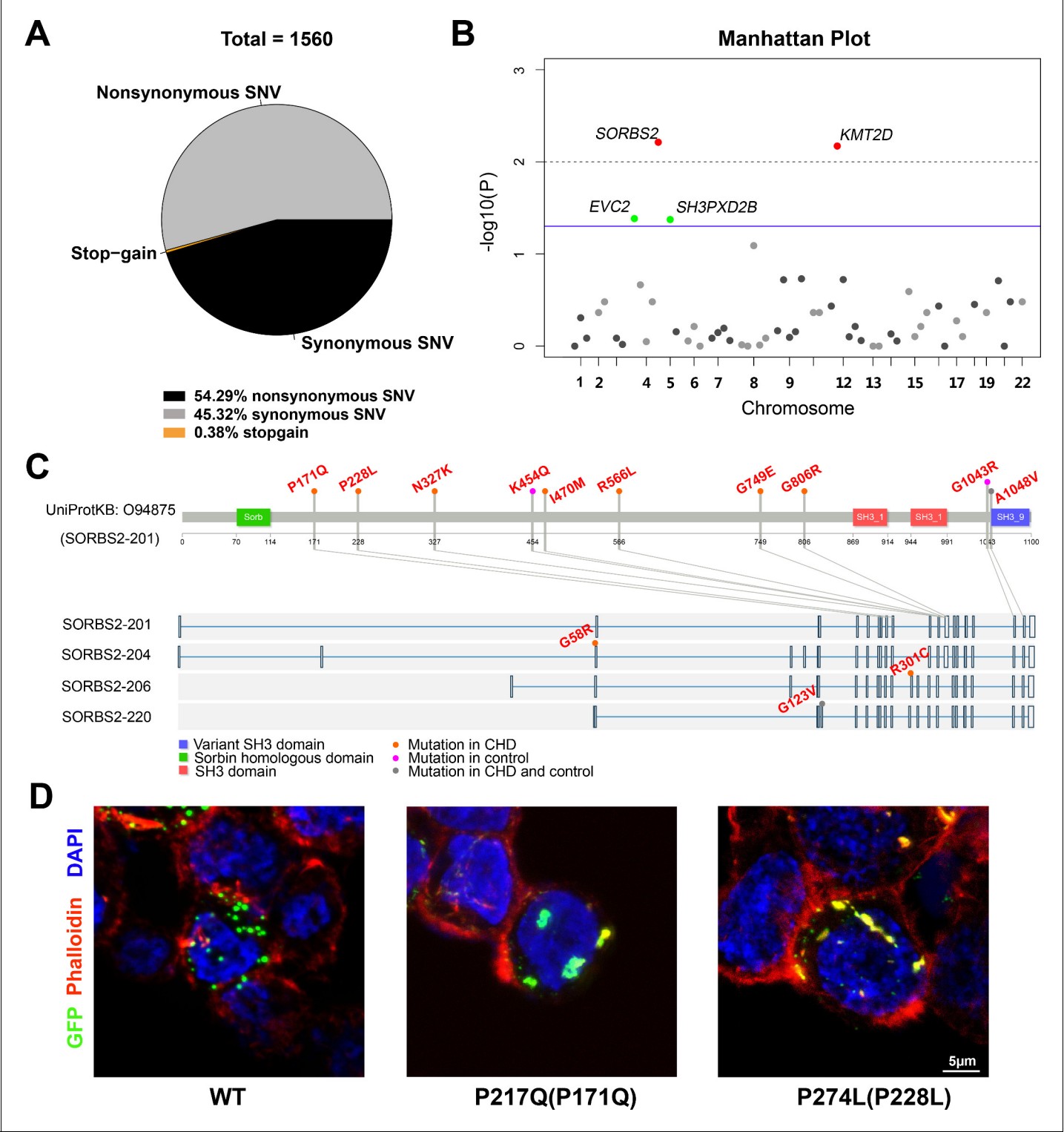

**Figure 4.** Rare *SORBS2* variants are enriched in CHD patients. (**A**) Descriptive statistics of the identified exonic variants. (**B**) Manhattan plot of gene-level Fisher's exact test of rare damaging variant counts between congenital heart disease (CHD) and control groups. Raw p-values of 0.05 and 0.01 are indicated by a blue line and a grey dash line, respectively. Genes (*SORBS2*, *KMT2D*) with a q-value lower than 0.2 are highlighted in red. Genes (*EVC2*, *SH3PXD2B*) with p<0.05 but q>0.2 are highlighted in green. (**C**) Illustration of rare damaging variants in *SORBS2*. Most variants are indicated in the longest *SORBS2* isoform (SORBS2-201). Three isoform-specific variants are shown in the corresponding exons. Variants in CHD and control groups are indicated by orange and pink dots, respectively. Variants appearing in both groups are indicated by grey dots. (**D**) Representative images of

*Figure 4 continued on next page*

*Figure 4 continued*

immunofluorescent staining of HEK293 cells transfected with EGFP-tagged SORBS2 (isoform 206) or variants. Amino acid coding in the bracket is the sequence numbering of isoform 201. Red, phalloidin staining.

The online version of this article includes the following figure supplement(s) for figure 4:

**Figure supplement 1.** Ethnic background comparison of CHD and control groups.

SHF (*Kelly, 2012*). Indeed, our data revealed that *SORBS2* functions not only as a sarcomeric component to maintain cardiomyocyte function, but also as an adaptor protein to promote SHF progenitor commitment in in vitro cardiomyocyte differentiation. ASD is detected in *Sorbs2⁻/⁻* mouse hearts. It supports that *Sorbs2* regulates SHF development in vivo and its role is conserved across species. In both models, we detected an increased protein level of NOTCH1 endocytosis facilitator c-ABL, a decreased NOTCH1 protein level, and impaired SHH signaling. Notch and Shh signaling is essential for SHF development (*Paige et al., 2015*). Notch signaling promotes Smo accumulation in cilia and enhances cellular response to Shh, placing Notch upstream of Shh signaling (*Stasiulewicz et al., 2015*; *Kong et al., 2015*). Therefore, *SORBS2* might promote SHF progenitor fate through c-ABL/ NOTCH/SHH axis. In addition, *Tbx5* was also significantly down-regulated in posterior SHF of *Sorb2* mutants. *Tbx5-Hh* molecular network is an essential regulatory mechanism in SHF for atrial septation (*Xie et al., 2012*). It is likely that *Tbx5* downregulation also contributes to the pathogenesis of ASD through its effect on Hh signaling. Consistently, adding recombinant SHH protein is sufficient to rescue SHF marker gene expression and cardiomyocyte differentiation efficiency. However, *TBX5* was up-regulated in D5 *SORBS2*-knockdown cells. It is likely that in vitro cardiomyocyte differentiation is a simplified model that cannot fully recapitulate the in vivo spatial and temporal information of various cardiac progenitors and, therefore, has less regulatory layers in which *TBX5* may predominantly function as an FHF regulator. Indeed, *Tbx5* knockdown reduces FHF progenitors and has no effect on SHF progenitors in an in vitro cardiogenesis model (*Andersen et al., 2018*).

Unlike human 4q deletion patients, *Sorbs2⁻/⁻* mice have no conotruncal defect and the penetrance of ASD is only 40%, indicating a relatively small effect of *SORBS2* in CHD pathogenesis. The high penetrance of conotruncal defects in human 4q deletion patients may be due to genetic modifiers that, together with *SORBS2* haploinsufficiency, cause the developmental defect in cardiac outflow tract. An obvious genetic modifier is the *HAND2* gene, which is co-missing with *SORBS2* in large 4q deletions. Previous studies have shown that CHD is observed more frequently in patients with the terminal deletion at 4q31 than in patients with the terminal deletion at 4q34 or 4q35 (*Lin et al., 1988*). Since some terminal 4q deletions that do not cover *HAND2* still manifest conotruncal defects, there may be another genetic modifier in the terminal deletion region. *Helt,* whose human homologous *HELT* is located within 4q35.1, encodes a Hey-related bHLH transcription factor that is expressed in both the brain and heart, and mediates Notch signaling (*Nakatani et al., 2004*). Therefore, both *SORBS2* and *HELT* haploinsufficiency might synergistically impair NOTCH signaling and cause a cardiac outlfow tract (OFT) defect.

DAS, also called Cor triatriatum type C in the original report (*Thilenius et al., 1976*), is a very rare CHD characterized by an extra septal structure to the right side of primary atrial septum (*Roberson et al., 2006*). This anatomic abnormality is implicated as a cause of paradoxical thromboembolic event to stroke or heart attack (*Breithardt et al., 2006*). However, its etiology and pathogenesis are entirely unknown. Here, we have shown that *Sorbs2* deficiency can cause this abnormality. Interestingly, Cor triatriatum, another type of abnormal extra atrial septation, has been reported in a patient with a terminal 4q34.3 deletion (*Marcì et al., 2015*), which includes *SORBS2* but not *HAND2*. It has been speculated that DAS might result from the persistence of embryologic structures or abnormal duplication of atrial septum. The impaired cardiogenesis in *Sorbs2⁻/⁻* mice suggests that the latter scenario may be the underlying pathogenesis.

# Materials and methods

**Key resources table**

| Reagent type (species) or resource | Designation | Source or reference | Identifiers | Additional information |
|---|---|---|---|---|
| Antibody | Anti-c-ABL (rabbit polyclonal) | Abclonal | A0282 | (1:1000) RRID:AB_2757094 |
| Antibody | Anti-Notch1 (rabbit monoclonal) | Cell Signaling Technology | 3608 s | (1:1000) RRID:AB_2153354 |
| Antibody | Anti-GAPDH (mouse monoclonal) | Abcam | ab8245 | (1:1000) RRID:AB_2107448 |
| Antibody | Anti-Rabbit IgG (HRP) (goat polyclonal) | Abcam | ab6721 | (1:5000) RRID:AB_955447 |
| Antibody | Anti-Mouse IgG (HRP) (goat polyclonal) | Abcam | ab205719 | (1:5000) RRID:AB_2755049 |
| Antibody | Anti-TRA-1–60 (mouse monoclonal) | Abcam | ab16288 | (1:200) RRID:AB_778563 |
| Antibody | Anti-Oct4 (rabbit polyclonal) | Abcam | ab18976 | (1:200) RRID:AB_444714 |
| Antibody | Anti-SOX2 (rabbit monoclonal) | Abcam | ab92494 | (1:200) RRID:AB_10585428 |
| Antibody | Anti-Cardiac Troponin I (mouse monoclonal) | Abcam | aab92408 | (1:200) RRID:AB_10562928 |
| Antibody | Anti-$\alpha$-Actinin (mouse monoclonal) | Sigma | A5044 | (1:200) RRID:AB_476737 |
| Antibody | Anti-mouse IgG, Alexa Fluor 488 (goat polyclonal) | Invitrogen | A11029 | (1:1000) RRID:AB_138404 |
| Antibody | Anti-rabbit IgG, AlexaFluor 633 (goat polyclonal) | Invitrogen | A21071 | (1:1000) RRID:AB_2535732 |
| Antibody | Anti-Cardiac Troponin T (mouse monoclonal) | Abcam | ab8295 | (1:200) RRID:AB_306445 |
| Antibody | FITC Anti-Cardiac Troponin T (mouse monoclonal) | Abcam | ab105439 | (1:100) RRID:AB_10866306 |
| Transfected construct (human) | Lentivirus: SORBS2-shRNA-psPAX2-pMD2.G | Addgene | psPAX2 (12260, Addgene) pMD2.G (12259, Addgene) | Lentiviral construct to transfect and express the shRNA |
| Cell line (*Homo sapiens*) | H1 hESC line | This paper | H1 hESC line-P21 | Provided by Chen's lab in Shanghai Institute of Biochemistry and Cell Biology (RRID:CVCL_9771) |

*Continued on next page*

*Continued*

| Reagent type (species) or resource | Designation | Source or reference | Identifiers | Additional information |
|---|---|---|---|---|
| Cell line (*Homo sapiens*) | ShRNA-SORBS2-H1-hESC | This paper | | Generated in Zhang's lab from Shanghai children's medical center |
| Software, algorithm | Clampfit 10.5/ Origin 8.0 | OriginLab, Northampton, MA, USA | | RRID:SCR_014212 |
| Software, algorithm | Image J | *Schneider et al., 2012* | https://imagej.nih.gov/ij/ | RRID:SCR_003070 |
| Software, algorithm | R | R Core Team, 2014 | https://www.r-project.org/ | RRID:SCR_001905 |
| Software, algorithm | Burrows-Wheeler Aligner | *Li and Durbin, 2009* | v0.7.17 | RRID:SCR_010910 |
| Software, algorithm | Picard Tools | Broad Institute | v2.21.8 | RRID:SCR_006525 |
| Software, algorithm | GATK | Broad Institute | v3.8 | RRID:SCR_001876 |
| Software, algorithm | Samtools | *Li and Durbin, 2009* | v1.9 | RRID:SCR_002105 |
| Software, algorithm | Annovar | *Wang et al., 2010* | v2019Oct24 | RRID:SCR_012821 |
| Biological sample (*Homo sapiens*) | Peripheral blood | This paper | | Isolated from of 300 children with complex CHD from Shanghai Children's Medical Center |
| Sequence-based reagent | Primers for RT-PCR | This paper | | Sequences are provided in *Supplementary file 13* |
| Sequence-based reagent | SORBS2-shRNA plasmid vectors (U6-MCS-Ubiquitin-Cherry-IRES-puromycin) | Shanghai Genechem Co | GIEE0117834 | shRNA-1 and shRNA-2 are 5'-TCCTTGTAT CAGTCCTCTA-3' and 5'-TCGATTCCACA GACACATA-3', respectively |
| Sequence-based reagent | In situ probe for Mouse *Tbx5* | This paper | | Provided by Dr. Lo's lab in University of Pittsburgh |
| Sequence-based reagent | In situ probe for Mouse *Heyl* | This paper | | Generated in house. Primers: F-5' GCCAGG AGCATAGTCCCAAT, R-5' GGCCCTCAAC CCACTCCATGAC |
| Sequence-based reagent | In situ probe for Mouse *Hoxb1* | This paper | | Generated in house. Primers: F—5' TTCCTTT TTAGAGTACCC ACTTTG, R-5' GTTTCTCTTGAC CTTCATCCAGTC |
| Commercial assay or kit | Illumina Genome Analyzer IIx platform | Illumina | | |
| Commercial assay or kit | Agilent SureSelect Capture panel | Agilent | | |

*Continued on next page*

*Continued*

| Reagent type (species) or resource | Designation | Source or reference | Identifiers | Additional information |
| --- | --- | --- | --- | --- |
| Commercial assay or kit | Reverse Transcription Kit | Takara | RR037A | |
| Commercial assay or kit | SYBR Fast qPCR Mix | Takara | RR430A | |
| Commercial assay or kit | TUNEL staining | Yeasen Biotech | T18120 | |
| Commercial assay or kit | Accutase | Stem cell Technologies | 7920 | |
| Commercial assay or kit | TRizol reagent | Thermo Fisher Scientific | 15596018 | |
| Commercial assay or kit | OCT | Thermo Fisher Scientific | 6502 | |
| Commercial assay or kit | Matrigel | BD Biosciences | 354277 | |
| Commercial assay or kit | TeSR-E8 medium | Stem cell Technologies | 05840 | |
| Commercial assay or kit | RPMI 1640 | Gibco | C14065500 | |
| Commercial assay or kit | L-ascorbic acid 2-phosphate | Sigma | 113170-55-1 | 213 µg/ml |
| Commercial assay or kit | Oryza sativa-derived recombinant human albumin | Healthgen Biotechnology Corp | HY100M1 | 500 µg/ml |
| Commercial assay or kit | CHIR99021 | Stem cell Technologies | 72052 | 6 µM |
| Commercial assay or kit | Wnt-C59 | Peprotech Biogems | 1248913 | 2 µM |
| Commercial assay or kit | Recombinant SHH protein | Sinobiological | 10372-H08H1 | |
| Commercial assay or kit | RIPA buffer | Beyotime | P0013B | |
| Other | B6.C-Tg(CMV-cre)1Cgn/J mice/C57 | This paper | | Jackson lab (RRID:IMSR_JAX:006054) |
| Other | Sorbs2 flox/flox mice/C57 | This paper | | Gifts from Dr. Guoping Feng's lab (RRID:IMSR_JAX:028600) |

## Mouse lines and breeding

Mice were housed under specific pathogen-free conditions at the animal facility of Shanghai Children's Medical Center. *Sorbs2^flox/flox* mice (*Zhang et al., 2016*) were a gift from Dr. Guoping Feng's lab (McGovern Institute for Brain Research, MIT, Cambridge). *Sorbs2^-* allele was obtained by breeding *Sorbs2^flox* allele into CMV-Cre mouse (*Schwenk et al., 1995*). The strains were backcrossed with C57BL/6 to maintain the lines ever since we obtained them. 2- to 6-month-old males and females were used for timed mating and embryos were collected at E18.5. Neither anesthetic nor analgesic agent was applied. $CO_2$ gas in a closed chamber was used for euthanasia of pregnant dam and cervical dislocation was followed. Isolated fetuses were euthanized by cervical dislocation. Animal care and use were in accordance with the NIH guidelines for the Care and Use of Laboratory Animals and approved by the Institutional Animal Care and Use Committee of Shanghai Children's Medical Center (SCMC-LAWEC-2017006).

## Histological analysis

For cardiac phenotype analysis, embryos were collected at E18.5 and E10.5, and were fixed in 10% formalin overnight. Isolated hearts were processed for paraffin embedding, sectioned at a thickness of 4 μm, and stained with hematoxylin and eosin. Stained sections were imaged using a Leica DM6000 microscope.

## RNA in situ hybridization

E10.5 embryos were collected and fixed in 4% paraformaldehyde (PFA) solution for 2 hr at room temperature, then dehydrated by 30% sucrose, and embedded in OCT (Thermo Fisher Scientific, 6502). 10 μm cryosections were used for RNA in situ hybridization (ISH) according to standard procedure. *Tbx5* probe plasmid was a gift of Dr. Cecilia Lo (University of Pittsburgh). Hoxb1 and Heyl probes were generated in house through PCR amplification. Primers for *Hoxb1* probe: F-5′ TTCCTTTTTAGAGTACCCACTTTG, R-5′ GTTTCTCTTGACCTTCATCCAGTC. Primers for *Heyl* probe: F-5′ GCCAGGAGCATAGTCCCAAT, R-5′ GGCCCTCAACCCACTCCATGAC.

## H1 hESC cell cultures and cardiomyocyte differentiation

H1 hESC line (gift of Dr. Xin Cheng, Shanghai Institute of Biochemistry and Cell Biology) was tested negative for mycoplasma with PCR assay. Undifferentiated H1 hESC lines were maintained in a feeder-free culture system. Briefly, we precoated the well plates with Matrigel (354277; BD Biosciences), and then seeded and cultured cells with TeSR-E8 medium (05840; Stemcell). When cells reached 80% confluence, they were passaged routinely with Accutase (07920; Stemcell). For cardiomyocyte differentiation, cells were induced using a chemically defined medium consisting of three components (CDM3): the basal medium RPMI 1640 (C14065500; Gibco), L-ascorbic acid 2-phosphate (213 μg/ml, 113170-55-1; Sigma), and Oryza sativa-derived recombinant human albumin (500 μg/ml, HY100M1; Healthgen Biotechnology Corp). In brief, single-cell suspensions were prepared using Accutase and were seeded in 12-well Matrigel-coated plate at a density of $4 \times 10^5$ cells/well. When cells reached 80–90% confluence (day 0), cells were fed by 2 ml CDM3 basal medium supplemented with CHIR99021 (6 μM, 72052; Stem cell). 48 hr later (day 2), the medium was replaced with 2 ml CDM3 supplemented with Wnt-C59 (2 μM, 1248913; Peprotech Biogems). After 96 hr (day 4), the medium was replaced with CDM3 basal medium every other day until the appearance of cell beating. In the rescue experiments, 250 μg/ml recombinant Shh protein (10372-H08H1; Sinobiological) was added in shRNA-*SORBS2* H1 hESCs at the beginning of D5 (when the medium was replaced with CDM3 basal medium). After 16 hr, some cells were collected for qRT-PCR. Others were left for immunofluorescent staining at D15.

Lentiviral shRNA plasmid vectors (U6-MCS-Ubiquitin-Cherry-IRES-puromycin) expressing target-specific sequences against human *SORBS2* and non-target scrambled shRNA were purchased from Shanghai Genechem Co. The targeting sequences of *SORBS2* shRNA-1 and shRNA-2 are 5′-TCCTTG TATCAGTCCTCTA-3′ and 5′-TCGATTCCACAGACACATA-3′, respectively. The sequence of scrambled shRNA is 5′-TTCTCCGAACGTGTCACGT-3′. Lentiviral particles were produced by transfecting human embryonic kidney (HEK) 293FT cells with shRNA, psPAX2 (12260; Addgene), and pMD2.G (12259; Addgene) plasmids. Efficiency of gene knockdown was examined using qRT-PCR.

## Western blot

Cells or embryos were lysed in radioimmunoprecipitation assay buffer (P0013B; Beyotime) containing protease inhibitors (P1010; Beyotime). Protein concentrations were determined with the BCA protein assay kit (Thermo). Protein was separated via 8% sodium dodecyl sulphate–polyacrylamide gel electrophoresis and, afterwards, transferred to a polyvinylidene difluoride membrane. After blocking by 5% non-fat milk for 1 hr, primary antibodies were incubated overnight at 4℃. The membrane was washed with tris-buffered saline with Tween-20 and incubated with secondary antibodies for 0.5 hr at room temperature. Bands were detected with the Immobilon ECL Ultra Western HRP Substrate (WBULS0500; Sigma) and band intensity was analyzed by ImageJ software. Antibodies: c-ABL (1:1000, A0282; Abclonal), NOTCH1 (1:1000, 3608 s; CST), and GAPDH (1:1000, ab8245; Abcam). Goat anti-rabbit IgG H and L (HRP) (1:5000, ab6721; Abcam), goat anti-mouse IgG H and L (HRP) (1:5000, ab205719; Abcam).

## Immunofluorescent staining

Cells were fixed in 4% PFA for 10 min, permeabilized in 0.5% Triton X-100/phosphate-buffered saline (PBS) for 20 min, and then blocked in 5% bovine serum albumin/PBS for 30 min. Fixed cells were stained with the following primary antibodies: TRI-1–60 (1:200, ab16288; Abcam), OCT4 (1:200, ab18976; Abcam), SOX2 (1:200, ab92494; Abcam), cTnI (1:200, ab92408; Abcam), cTnT (1:200, ab8295; Abcam), and α-actinin (1:200, A5044; Sigma). These primary antibodies were visualized with AlexaFluor 488 (1:1000, A11029; Invitrogen) or AlexaFluor 633 (1:1000, A21071; Invitrogen). TUNEL staining was performed with a commercial kit according to manufactory menu (T18120; Yeasen Biotech). GFP-SORBS2 plasmids were transfected into HEK293 cells. After 48 hr, cells were fixed with 4% PFA for 20 min and treated with 0.1% triton X-100 for 10 min. F-actin was stained with Acti-stainTM 555 Fluorescent Phalloidin (Cat. #PHDH1; Cytoskeleton). Nuclei were stained with 4′,6-diamidino-2-phenylindole. Fluorescent images were acquired using a Laser confocal microscope (Leica TCS SP8).

## Flow cytometric analysis

In brief, D15 cells were harvested in 0.25% trypsin/EDTA at 37°C for 15 min and subsequently neutralized by 10% fetal bovine serum in Dulbecco's modified Eagle medium. Then cells were centrifuged at 1000 rpm for 5 min and resuspended in Invitrogen FIX and PERM solution and kept at 4°C for 30 min. After washing, cells were incubated with anti-cTnT antibody (1:100, ab105439; Abcam) in washing buffer on ice in the dark for 45 min. Cells were centrifuged, washed, and resuspended for detection. Data were collected by BD FACSCanto flow cytometer and analyzed by BD FACS software.

## Electron microscopy

D30 cardiomyocytes were harvested using 0.25% trypsin/EDTA and prefixed with 2.5% glutaraldehyde in 0.2 M phosphate buffer overnight at 4°C. Samples were washed and then post-fixed with 1% osmium tetroxide for 1.5 hr. Next, cells were routinely dehydrated in an ethanol series of 30, 50, 70, 80, and 95% for 15 min each, and 100% ethanol and acetone twice for 20 min each at room temperature, and then embedded in an epoxy resin. Sections (70 nm thick) were poststained in uranyl acetate and lead citrate and visualized on Hitachi 7650 microscope.

## Electrophysiological recordings

D30 h1ESC-CMs were digested by Accutase (7920; STEMCELL Technologies, Canada), washed one time with a baseline extracellular fluid, and then moved to the stage of an inverted microscope (ECLIPSE Ti-U; Nikon, Japan) for patch-clamp recording. h1ESC-CMs were continuously perfused by an extracellular solution through a 'Y-tube' system with a solution exchange time of 1 min. Whole-cell patch-clamp recordings were performed using Axopatch 700B (Axon Instruments, Inc, Union City, CA, USA) amplifiers under an invert microscope at room temperature (22–25°C). Glass pipettes were prepared using borosilicate glasses with a filament (Sutter Instruments Co, Novato, CA) using the Flaming/Brown micropipette puller P97 (Sutter Instruments Co). The final resistance parameters of patch pipette tips were about 2–4 MΩ after heat polish and internal solution filling. After the formation of 'gigaseal' between the patch pipette and cell membranes, a gentle suction was operated to rupture the cell membrane and establish whole-cell configuration. All current signals were digitized with a sampling rate of 10 kHz and filtered at a cutoff frequency of 2 kHz (Digidata 1550A; Axon Instruments, Inc, Union City, CA). The spontaneous action potentials were recorded in a gap-free mode with a sampling rate of 1 kHz and filtered at a cutoff frequency of 0.5 kHz. If the series resistance was more than 10 MΩ or changed significantly during the experiments, the recordings were discarded from further analyses. The pipette internal solution for action potential recording contained (in mM) KCl 150, NaCl 5, $CaCl_2$ 2, EGTA 5, HEPES 10, and MgATP 5 (pH 7.2, KOH), and baseline extracellular solution and extracellular solution for action potential recording contained (in mM) NaCl 140, KCl 5, $CaCl_2$ 1, $MgCl_2$ 1, glucose 10, and HEPES 10 (pH 7.4, NaOH). Data were analyzed by using Clampfit 10.5 and Origin 8.0 (OriginLab, Northampton, MA).

## Quantitative real-time PCR analysis

Total RNA was extracted from D0, D2, D3, D5, and D10 cells or E10.5 embryos using Trizol reagent (15596018; Thermo Fisher Scientific). Reverse transcription was accomplished with Reverse Transcription Kit (Takara; RR037A) according to the manufacturer's instructions. qPCR was performed with the SYBR Fast qPCR Mix (Takara; RR430A) in the Applied Biosystems 7900 Real-Time PCR System. Primers are listed in *Supplementary file 10*.

## RNA-Seq

Total RNA of D5 cells and E10.5 embryos was isolated using TRizol reagent (Thermo Fisher Scientific; 15596018). Library preparation and transcriptome sequencing on an Illumina HiSeq platform were performed by Novogene Bioinformatics Technology Co, Ltd to generate 100-bp paired-end reads. HTSeq v0.6.0 was used to count the read numbers mapped to each gene, and fragments per kilobase of transcript per million fragments mapped (FPKM) of each gene were calculated. We used FastQC to control the quality of transcriptome sequencing data. Next, we compared the sequencing data to the human reference genome (hg19) by STAR. The expression level of each gene under different treatment conditions is obtained by HTSeq-count after standardization. The differentially expressed genes were analyzed by DESeq2 package. Functional enrichment of differentially expressed genes was analyzed on Toppgene website.

GSE126128 and GSE131181 datasets were retrieved from Gene Expression Omnibus (GEO) database. Seurat (version 3.0) toolkit was used for scRNA-seq analysis. After data integration, batch effect elimination, normalization, and scaling, different cell populations were identified based on existing references. Gene expression was plotted using normalized read counts.

## Network and GO analysis

From ENCODE database (*Gerstein et al., 2012*), 316 human fetal heart-specific genes including *SORBS2* were selected and their expression coefficients were computed. The highly co-regulated transcriptional networks (correlation coefficient ≥0.8) were constructed and visualized with BioLayout Express3D. The interconnected gene clusters were detected using the MCL (Markov Cluster) algorithm and illustrated with different colors.

## Patient samples

A total of 300 children with complex CHD, including ASD, conotruncal defect, and so on, were enrolled in our study from November, 2011 to January, 2014 in Shanghai Children's Medical Center (*Supplementary file 11*). Patients carrying 22q11.2 deletion and gross chromosomal aberrations were excluded from our study. The mean age of included probands was 10 months with a range of 3 days to 17 years. 188 (62.7%) were boys and 112 (37.3%) were girls. CHD diagnosis was confirmed by reviewing patient history, physical examinations, and medical records. Patients carrying 22q11.2 deletion and gross chromosomal aberrations (e.g., trisomy 21, trisomy 13, and trisomy 18) were excluded from our study. The study conformed to the principles outlined in the Declaration of Helsinki, and approval for human subject research was obtained from the Institutional Review Board of Shanghai Children's Medical Center (SCMC-201015). Written informed consents were obtained from parents or legal guardians of all patients.

## Control cohort

Exome sequences for 220 control subjects of Han Chinese descent were derived from the 1000 genome project (http://www.internationalgenome.org/data). Raw sequence data in the form of fastq files were re-aligned and re-analyzed with the same bioinformatic pipelines with CHD patients. The ethnicity of cases and controls was investigated by performing PCA with single nucleotide polymorphism (SNP) genotype data from all the participants of this study.

## Gene selection and targeted sequencing

We used a customized capture panel of 104 targeted genes, which included 81 known CHD genes (*Supplementary file 12*) and 23 CHD candidate genes (*Supplementary file 13*). The CHD genes were selected through a comprehensive literature search and had been reported by other research groups to be associated with CHD in either human patients or mouse models (*Andersen et al.,*

2014; *Barriot et al., 2010*; *Fahed et al., 2013*). CHD candidate genes were prioritized from pathogenic CNVs and likely pathogenic CNVs identified in CHD patients in our previous study (*Geng et al., 2014*). The coding regions of selected genes and their flanking sequences were covered by the Agilent SureSelect Capture panel and sequenced on the Illumina Genome Analyzer IIx platform according to the protocols recommended by the manufacturers.

## Variant calling and quality control

We used the best practice pipeline of Broad Institute's genome analysis toolkit (GATK) 3.7 to obtain genetic variants from the target sequencing data of 300 CHD cases together with the raw data of 220 Han Chinese control samples. Briefly, Burrows-Wheeler Aligner (BWA, version 0.7.17) was used to align the Fastq format sequences to human genome reference (hg38). De-duplication was performed using Picard, and the base quality score recalibration (BQSR) was performed to generate analysis-ready reads. HaplotypeCaller implemented in GATK was used for variant calling in genomic variant call format (GVCF) mode. All samples were then genotyped jointly. We then excluded the variants based on the following rules: (1) >2 alternative alleles; (2) low genotype call rate <90%; (3) deviation from Hardy-Weinberg Equilibrium in control samples ($p < 10^{-7}$); (4) differential missingness between cases and controls ($p < 10^{-6}$). After alignment and variant calling, we removed two subjects with low-quality data, leaving a total of 298 cases and 220 controls.

## Variant enrichment analysis

Given that severe mutations are generally present at low frequencies in the population, we set the relevant variants filtering criteria as follows: (1) variants that located in exonic region, (2) excluding synonymous variants, (3) variants with an MAF below 1% according to the public control database (Exome Aggregation Consortium, ExAC), and (4) damaging missense variants predicted to be deleterious by at least two algorithms (Polyphen2 $\geq 0.95$/MutationTaster_pred:D/SIFT $\leq 0.05$). All relevant variants following these criteria were hereafter called 'rare damaging' variants. The number of rare damaging variant carriers in each gene was counted in CHD patients and controls. We hypothesized that rare damaging variants should be enriched in CHD patients, hence the carrier and non-carrier groups were compared between CHD patients and controls using a one-tailed Fisher's exact test. The odds ratio (OR) was calculated. Only genes with at least two variants were retained, and multiple testing correction was performed using Benjamini-Hochberg procedure (q-value, adjusted p-value after Benjamini-Hochberg testing).

## Statistical analysis

Statistical significance was performed using a two-tailed Student's *t* test, $\chi^2$ test, or Fisher's exact test as appropriate. The combined difference of Notch and Shh signaling was tested by non-parametric PERMANOVA. Statistical significance is indicated by *$p < 0.05$ and **$p < 0.01$.

# Acknowledgements

We thank Dr. Guoping Feng at Massachusetts Institute of Technology for *Sorbs2^flox/flo*x mice, Dr. Xin Cheng at Shanghai Institute of Biochemistry and Cell Biology for H1 ES cell line, and Dr. Bingshan Li at Vanderbilt University and Dr. Hao Mei at University of Mississippi Medical Center for advice on genetic data analyses.

# Additional information

## Funding

| Funder | Grant reference number | Author |
|---|---|---|
| Science and Technology Commission of Shanghai Municipality | 20JC1418500 | Qihua Fu |
| Collaborative Innovation Program of Shanghai Municipal Health Commission | 2020CXJQ01 | Zhen Zhang |

| National Natural Science Foundation of China | 81371893 | Qihua Fu |
|---|---|---|
| National Natural Science Foundation of China | 31371465 | Zhen Zhang |
| Shanghai Municipal Education Commission-Gaofeng Clinical Medicine Grant Support | 20171925 | Zhen Zhang |
| Shanghai Sailing Program | 18YF1414800 | Xiaoqing Zhang |
| Innovative Research Team of High-level Local Universities in Shanghai | SSMU-ZDCX20180200 | Zhen Zhang |
| Science and Technology Commission of Shanghai Municipality | 20DZ2260900 | Qihua Fu |
| National Natural Science Foundation of China | 81741031 | Qihua Fu |
| National Natural Science Foundation of China | 81871717 | Qihua Fu |
| National Natural Science Foundation of China | 81672090 | Qihua Fu |
| National Natural Science Foundation of China | 31771612 | Zhen Zhang |

The funders had no role in study design, data collection and interpretation, or the decision to submit the work for publication.

## Author contributions

Fei Liang, Data curation, Formal analysis, Investigation, Methodology, Writing - original draft; Bo Wang, Data curation, Formal analysis, Investigation, Writing - original draft; Juan Geng, Data curation, Investigation; Guoling You, Min Zhang, Hunying Sun, Data curation, Formal analysis; Jingjing Fa, Investigation; Huiwen Chen, Resources; Qihua Fu, Conceptualization, Supervision, Funding acquisition, Writing - review and editing; Xiaoqing Zhang, Data curation, Formal analysis, Funding acquisition, Investigation, Writing - original draft; Zhen Zhang, Conceptualization, Supervision, Funding acquisition, Project administration, Writing - review and editing

## Author ORCIDs
Bo Wang ORCID https://orcid.org/0000-0003-4376-1398
Zhen Zhang ORCID https://orcid.org/0000-0002-9898-054X

## Ethics
Human subjects: The study conformed to the principles outlined in the Declaration of Helsinki and approval for human subject research was obtained from the Institutional Review Board of Shanghai Children's Medical Center (SCMC-201015). Written informed consents were obtained from parents or legal guardians of all patients.
Animal experimentation: Animal care and use were in accordance with the NIH guidelines for the Care and Use of Laboratory Animals and approved by the Institutional Animal Care and Use Committee of Shanghai Children's Medical Center (SCMC-LAWEC-2017006).

## Decision letter and Author response
Decision letter https://doi.org/10.7554/eLife.67481.sa1
Author response https://doi.org/10.7554/eLife.67481.sa2

# Additional files

## Supplementary files

- Supplementary file 1. Down-regulated genes in *SORBS2*-knockdown D5 cells for GO analysis.
- Supplementary file 2. Up-regulated genes in *SORBS2*-knockdown D5 cells for GO analysis.
- Supplementary file 3. Genotyping distribution in embryos from *Sorbs2*$^{+/-}$ mouse intercross.
- Supplementary file 4. Down-regulated genes in E10.5 *Sorbs2*$^{+/-}$ embryos for GO analysis.
- Supplementary file 5. Down-regulated genes in E10.5 *Sorbs2*$^{-/-}$ embryos for GO analysis.
- Supplementary file 6. Number of exonic variants detected in CHD and normal controls.
- Supplementary file 7. Carriers of rare damaging variants in CHD and normal controls.
- Supplementary file 8. CHD patients with rare *SORBS2* variants.
- Supplementary file 9. Cardiac phenotype in patients carrying *SORBS2* variants.
- Supplementary file 10. Primers for qPCR.
- Supplementary file 11. Subphenotypes of CHD cohort.
- Supplementary file 12. Known CHD genes included in targeted sequencing panel.
- Supplementary file 13. Candidate CHD genes included in targeted sequencing panel.
- Transparent reporting form

## Data availability

Targeted sequencing raw data of CHD patients have been deposited in NCBI's Sequence Read Archive (PRJNA579193). RNA-seq data have been deposited in NCBI's Gene Expression Omnibus (GSE137090).

The following datasets were generated:

| Author(s) | Year | Dataset title | Dataset URL | Database and Identifier |
|---|---|---|---|---|
| Fu Q | 2020 | Targeted sequencing of children with congenital heart disease | https://www.ncbi.nlm.nih.gov/sra/PRJNA579193 | NCBI Sequence Read Archive, PRJNA579193 |
| Liang F, Zhang X, Wang B, Zhang Z, Fu Q | 2021 | RNA-Sequencing analyses of control and SORBS2 knockdown cardiac progenitor cells derived from human stem cells in vitro and E10.5 wild-type and Sorbs2 knockout embryos | https://www.ncbi.nlm.nih.gov/geo/query/acc.cgi?acc=GSE137090 | NCBI Gene Expression Omnibus, GSE137090 |

The following previously published datasets were used:

| Author(s) | Year | Dataset title | Dataset URL | Database and Identifier |
|---|---|---|---|---|
| Hill MC | 2019 | A cellular atlas of Pitx2-dependent cardiac development | https://www.ncbi.nlm.nih.gov/geo/query/acc.cgi?acc=GSE131181 | NCBI Gene Expression Omnibus, GSE131181 |
| Soysa TY, Gifford CA, Srivastava D | 2019 | Single-cell analysis of cardiogenesis reveals basis for organ level developmental defects | https://www.ncbi.nlm.nih.gov/geo/query/acc.cgi?acc=GSE126128 | NCBI Gene Expression Omnibus, GSE126128 |

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
