## [Decision Letter]

**Acceptance summary:**

Fei Liang et al., have identified the deletion of the gene SORBS2 as a contributor to the congenital heart disease phenotype in the 4q deletion syndrome. They have shown that the gene is required for the development of the dorsal mesenchyme protrusion, a structure necessary for septation of the atria. To reach these conclusions, the team has used human genetics data, an in vitro differentiation model of human embryonic stem cells, and a mouse model of Sorbs2 loss of function. The data presented indicate that Sorbs2 is important for atrial septation by supporting Tbx5 expression as well as NOTCH and SHH signalling. The work supports a role for Sorbs2 in heart development and adds novel information to the genetics of 4q deletion syndrome.

**Decision letter after peer review:**

[Editors’ note: the authors submitted for reconsideration following the decision after peer review. What follows is the decision letter after the first round of review.]

Thank you for submitting your work entitled "SORBS2 is a genetic factor contributing to cardiac malformation of 4q deletion syndrome" for consideration by *eLife*. Your article has been reviewed by 3 peer reviewers, one of whom is a member of our Board of Reviewing Editors, and the evaluation has been overseen by a Senior Editor. The reviewers have opted to remain anonymous.

Our decision has been reached after consultation between the reviewers. Based on these discussions and the individual reviews below, we regret to inform you that your work will not be considered further for publication in *eLife*.

Your study addresses the genetic basis of an interesting, and as yet unclear human condition. The strength of your work is in the use of both mouse and human systems. However, the reviewers have underlined insufficient depth in the datasets from both systems, thus preventing a robust support for your conclusions.

Hoping to help you to increase the strength of your work, I summarize the issues that have been raised:

1. Mouse data need to be expanded significantly to better define the phenotype and to identify a developmental mechanism.

This work should also include validation of the proposed mechanism c-ABL/NOTCH/SHH.

2. Is there a haploinsufficiency phenotype in the mouse? While there may be species-specific differences in sensitivity to gene dosage, the authors should make an effort to investigate this question. There could be, for example a molecular phenotype (rather than a morphogenetic one) in the heterozygous mutant.

3. The human genetics data should be extended: a) correlate more directly variants and ASD; b) validate variants to demonstrate their functional significance.

*Reviewer #1:*

The manuscript describes the identification of the gene SORBS2 as a likely contributor to the congenital heart disease (CHD) phenotype in patients with 4q deletions. The support for the implication of the SORBS2 gene derives from the phenotype of *Sorbs2^-/-^* embryos, from human genetics studies, and from analysis of differentiated human ES cells in which this gene has been knocked down using siRNAs. The authors conclude that SORBS2 has at least two "functions" in the cardiogenic lineage. One is related to the sarcomere structure of the cardiomyocyte, a role to a certain extent predicted by previous studies of the SORBS2 protein, which is localized in the Z line and is a multi-domain scaffolding protein involved in cytoskeletal organization. The other, more novel role and more related to heart development, would be "… that SORBS2 is a critical regulator to maintain the balance between FHF and SHF cells during differentiation". The support for the latter role is much more speculative and requires more extensive studies, which would considerably increase the competitiveness of this manuscript.

– The cardiac development phenotype of *Sorbs2^-/-^* embryos should be investigated in greater detail, also with the aid of regional and lineage marker analyses, to corroborate the conclusions drawn by the hES cell culture experiments. Without these studies, the conclusion that Sorbs2 is involved in fate specification of SHF progenitors is not supported.

For example, RNA-seq data of differentiated hES cells suggest that knock-down of Sorbs2 reduces the expression of anterior SHF markers, such as Tbx1, and increases the expression of posterior SHF markers such as Tbx5, thus, the imbalance could be between aSHF and pSHF, rather than between FHF and SHF as suggested by the authors. Thus, is TBX5 expression altered in the pSHF of mutant embryos? How is the expression of other pSHF markers? Where is Sorbs2 expressed?

It is virtually impossible to predict mechanisms leading to cardiac structural defects using hES differentiation data. Even though the mouse phenotype is incompletely penetrant, it should still be possible to analyse it in more details and validate the proposed c-ABL/NOTCH/SHH pathway in vivo.

– Human genetics studies were performed on a relatively small number of patients; nevertheless they provided some suggestion that SORBS2 may have a role in CHD. The authors should show a map of the gene model or predicted protein with the location of the variants and list the actual variants that they found.

Also, did the 20 patients that carried "rare damaging variants" of SORBS2 also have variants in other CHD genes?

*Reviewer #2:*

In this manuscript, the authors present their study on the identification of SORBS2 as regulator of cardiac defects associated to Chr 4q deletion syndrome. The aim of this study was to identify the role of SORBS2, as a candidate gene for CHD of terminal 4q deletion syndrome, during cardiac development. The authors found that *Sorbs2^-/-^* mouse hearts have atrial septal defects (ASD) including rare double atrial septum. The authors used human ES cell lines as in vitro model to examine the requirement of SORBS2 during cardiomyocyte differentiation. Thus, they knockdown SORBS2 expression with 2 different shRNAs. Video shown that differentiated SORBS2-knockdown cardiomyocytes contracted much weaker than controls. Quantification of cTnT+ cells confirmed the later observation. Myofibril structure examination showed abnormal sarcomeric structure when SORBS2 expression is reduced whereas electrical activity seemed normal. The authors hypothesized that SORBS2 had an early role in cardiomyocyte differentiation. Expression of progenitor but not mesodermal markers was disturbed in SORBS2-knockdown hES cells. Based on expression analysis, the authors claimed that expression of first heart field (FHF) genes was increased whereas second heart field (SHF) markers were reduced. The authors concluded that SORBS2 is a critical regulator to maintain the balance between FHF and SHF cells. To understand how SORBS2 regulates SHF progenitor commitment, the authors performed RNA-seq experiments. RNA-seq analysis showed that the NOTCH signaling target genes were reduced in SORBS2-knockdown cardiomyocytes. Finally, in this study the authors used rare variant association to identify genes within CNVs responsible for CHD. Hence, the authors sequenced 300 complex CHD cases. Among genes with a p-value lower than 0.05, they found SORBS2. The authors identified missense mutations in SORBS2 suggesting that rare variants in this gene could be associated to ASD.

Although the basic insight regarding the identification of atrial septal defects in SORBS2 mutant mice and the reduction of cardiomyocyte differentiation in hES cells when SORBS2 is reduced are interesting, there are some weaknesses in the arguments and data reported in this study.

– My main concern is related to the arguments used to claim a role of SORBS2 in SHF development through c-ABL/NOTCH/SHH axis.

– It is clear that Sorbs2-null mice have ASD. However, the authors did not find the origin of this defect. The authors should show how dorsal mesenchymal protrusion is formed in these mutant embryos.

– The use of shRNA has some limitations that the authors did not discuss. Reduction of SORBS2 expression resulted in reduction of cTnT-positive cells. However, the authors should show whether apoptosis is normal in shRNA treated cells.

– The expression of sarcomeric genes is reduced in SORBS2-KD cells. How do the authors explain this result?

– In this study the qPCR is used to examine transcriptional levels of FHF and SHF genes during cardiomyocyte differentiation. Increase of TBX5, HAND1 and HCN4 is observed at D5 but clearly significant at D10. I'm wondering whether these markers signed the FHF or the atrial identity? The authors should validate this observation using other FHF markers.

– qPCR shows a decrease of three SHF genes. However, there is no SHF markers (except FGF8) listed in TableS2 (downregulated genes at D5 from RNA-seq analysis). The authors should discuss this point.

– I do not understand why the authors focused on the c-ABL/NOTCH/SHH axis to explain the role of SORBS2 in SHF development. The arguments that link NOTCH to SHH signaling pathways in this context are very weak. The mechanism is incomplete and here again the mouse model should have been used to validate these observations.

– The authors sequenced 300 CHD cases and found several missense variants in SORBS2 gene. TableS6 shows heart phenotype in patients. "The majority of patients with rare SORBS2 variants had ASD". The authors should show whether there is any correlation between SORBS2 mutation and ASD phenotype.

*Reviewer #3:*In this study, the Authors have attempted to use a mouse model to link the cardiac malformations observed in patients with the 4q deletion to defects in the SORBS2 locus. It is interesting to see that 40% of *Sorbs2^-/-^* hearts had ASD, and that some of them have an absence/hypoplasia of the primary septum and/or double atrial septum with an incomplete penetrance. However, while the ratio of cardiac defects in embryos is similar to the percentage of reported postnatal lethality, in general ASD hardly causes 100% postnatal lethality and likely cannot provide an adequate explanation as the sole cause of the early postnatal death. In addition, ASDs were only seen in the case of null genotypes (*Sorbs2^-/-^*), and the authors didn't report ASDs in the case of Sorbs2+/- genotypes. This is crucial because in 4q-, SORBS2 is going to be haploinsufficient, and not a null genotype.The major fault of the study is, if the authors feel that SORBS2+/- haploinsufficiency is the cause for cardiac defects, patient samples (from 4q- or SORBS2 VUCS) should be used to derive iPSCs instead of the hRNA knockdown ESCs for all subsequent studies.

SORBS2-knockdown iPSCs were used for cardiomyocyte differentiation, which cannot be directly linked to cardiac defects such as ASD. Since *Sorbs2^-/-^* is available, the molecular mechanisms should be tested with an in vivo model or primary cell cultures derived from *Sorbs2^-/-^* or Sorbs2+/- models.

Finally, while the Authors found that rare SORBS2 variants are significantly enriched in CHD patients, there is no evidence to support that these variants CAUSE cardiac defects.

[Editors’ note: further revisions were suggested prior to acceptance, as described below.]

Thank you for submitting your article "SORBS2 is a genetic factor contributing to cardiac malformation of 4q deletion syndrome" for consideration by *eLife*. Your article has been reviewed by 3 peer reviewers, one of whom is a member of our Board of Reviewing Editors, and the evaluation has been overseen by Didier Stainier as the Senior Editor. The following individual involved in review of your submission has agreed to reveal their identity: Stephane Zaffran (Reviewer #3).

Essential Revisions:

There are two sets of experiments that are deemed critical (please see details in the individual reviews):

1. A rescue experiment using recombinant NOTCH or SHH in the cell culture model.

2. Test additional pSHF markers by in situ hybridization on the DMP or pSHF of mutant embryos to validate and regionalize at least some of the expression changes revealed by whole-embryo RNA-seq.

*Reviewer #1:*

The resubmitted version has addressed the issues raised by the previous round of review, albeit some questions remain.

I have 3 recommendations for authors:

1. Phenotypic analysis in the mouse mutant has been extended, but it is still limited. However, the authors did find developmental abnormalities with the DMP, which is interesting (albeit predictable). Phenotyping would benefit from testing additional pSHF markers and perhaps by testing basic mechanisms such as cell proliferation and cell death

2. Mechanisms: the focus of the authors in a so called c-Abl-Notch1-Shh "axis" remains speculative and based on gene (and in some cases protein) expression assays in differentiated hES cells and in whole embryo RNA-seq. These results are insufficient to draw conclusions about the pSHF or whatever it is happening in the DMP. Claims to the effect that Sorbs2 specifies "the fate of second heart field (SHF) progenitors through c-ABL/NOTCH/SHH axis" are unsubstantiated and should be removed (in the abstract, subheading of results, discussion), or be proposed as a speculative hypothesis in the discussion. In addition, the authors show no evidence that Sorbs2 specifies the fate of SHF progenitors. On the other hand, new data added to the resubmission clearly show a downregulation of Tbx5 in the DMP/pSHF region of mutants. It is unclear whether this is due to hypoplasia of the DMP or to reduced expression of the gene. In any case, this could be a valuable clue to understand mechanisms as Tbx5 is a critical factor in DMP development (e.g. Xie et al., 2012, Moskowitz group, which has also shown interaction between Tbx5 and SHH signalling), thus it could be part of the pathogenesis of ASD in this mutant. This should be discussed and contrasted with the NOTCH hypothesis.

3. The authors should discuss the apparently inconsistent findings of increased expression of Tbx5 in the hES cell model, and the decreased expression of the same gene in the pSHF of mutant embryos.

*Reviewer #2:*

Authors have done an incredible job to revise the manuscript. The data is convincing that SORBS2 contributes to the cardiac phenotypes in 4q deletion syndrome.

It is interested to see the gene expression profile of heterozygous mutant is closer to that of homozygous mutants than to wild type.

Correlation of the SORBS2 variants with ASD and finding the enrichment of SORBS2 variants in patients with ASD added the value to the manuscript.

*Reviewer #3:*

The authors have replied to several of my concerns such as the arguments that SORBS2 specifies SHF progenitors through c-ABL/NOTCH/SHH. They used PCR, RNA-seq and in situ hybridization to validate their hypothesis. Thus, they showed that Notch1 protein expression level is significantly decreased whereas c-ABL is increased in *Sorbs2^-/-^* mutant embryos. In this revised version, they better used their in vivo model to support their hypothesis. The phenotype of the DMP is now examined which is consistent with ASD observed at E18.5. Control quality of shRNA KD hES cells is now performed and no apoptosis is detected. Expression of pSHF markers is now shown including Hoxb1, Hoxa1, Aldh1a2 and Tbx5 which confirms the previous statement. The rational to examine NOTCH and SHH signals is now better explained. Finally, human genetics data is improved and detection of significant enrichment of SORBS2 rare damaging variant in patients with ASD vs non-ASD is great. However, I agree with reviewer#3 that hiPS cells model rather than hESC would have been better to support their hypothesis. In addition, a rescue experiment using recombinant NOTCH or SHH treatment in the in vitro culture would have been important to perform in order to support the main finding of this study.

---

## [Author Response]

[Editors’ note: the authors resubmitted a revised version of the paper for consideration. What follows is the authors’ response to the first round of review.]

Your study addresses the genetic basis of an interesting, and as yet unclear human condition. The strength of your work is in the use of both mouse and human systems. However, the reviewers have underlined insufficient depth in the datasets from both systems, thus preventing a robust support for your conclusions.Hoping to help you to increase the strength of your work, I summarize the issues that have been raised:1. Mouse data need to be expanded significantly to better define the phenotype and to identify a developmental mechanism.This work should also include validation of the proposed mechanism c-ABL/NOTCH/SHH.

As per reviewers’ suggestion, we have analyzed early embryos and identified two types of DMP malformations that matches with E18.5 phenotypes. We have also validated our in vitro molecular findings in mouse embryos. Please check details in the following responses to reviewers.

2. Is there a haploinsufficiency phenotype in the mouse? While there may be species-specific differences in sensitivity to gene dosage, the authors should make an effort to investigate this question. There could be, for example a molecular phenotype (rather than a morphogenetic one) in the heterozygous mutant.

We have used RNA-seq to check transcriptome of wild type, heterozygous and homozygous mutants. Although heterozygous mutant is phenotypically similar to wild type, the gene expression profile of heterozygous mutant is more similar to that of homozygous mutant. GO analysis showed that the affected biological processes and pathways are also similar to those affected in homozygous mutant

3. The human genetics data should be extended: a) correlate more directly variants and ASD; b) validate variants to demonstrate their functional significance.

We have checked *SORBS2* frequency difference between patients with ASD and without ASD in our cohort. Indeed, we noted a significant enrichment of *SORBS2* variants in patients with ASD. We also over-expressed two of *SORBS2* variants in HEK293 cells and found these mutations could alter cellular distribution of SORBS2 protein. Please check details in the following responses to reviewers.

Reviewer #1:The manuscript describes the identification of the gene SORBS2 as a likely contributor to the congenital heart disease (CHD) phenotype in patients with 4q deletions. The support for the implication of the SORBS2 gene derives from the phenotype of ^-/-^Sorbs2^-/-^ embryos, from human genetics studies, and from analysis of differentiated human ES cells in which this gene has been knocked down using siRNAs. The authors conclude that SORBS2 has at least two "functions" in the cardiogenic lineage. One is related to the sarcomere structure of the cardiomyocyte, a role to a certain extent predicted by previous studies of the SORBS2 protein, which is localized in the Z line and is a multi-domain scaffolding protein involved in cytoskeletal organization. The other, more novel role and more related to heart development, would be "… that SORBS2 is a critical regulator to maintain the balance between FHF and SHF cells during differentiation". The support for the latter role is much more speculative and requires more extensive studies, which would considerably increase the competitiveness of this manuscript.

*TBX5* is generally considered as a FHF marker in in vitro cardiomyocyte differentiation model. In our in vitro differentiation data, we observed up-regulation of several FHF marker genes, including *TBX5*. As the reviewer mentioned, Tbx5 is also considered as a posterior SHF marker in in vivo context. In regarding to the mouse phenotype, it is more important to establish the role of *Sorbs2* in promoting SHF development rather than a role in maintaining the balance between FHF and SHF. Therefore, we have removed this hypothetical statement in the text. We respectfully request reviewers and editors not to ask these data.

– The cardiac development phenotype of ^-/-^Sorbs2^-/-^ embryos should be investigated in greater detail, also with the aid of regional and lineage marker analyses, to corroborate the conclusions drawn by the hES cell culture experiments. Without these studies, the conclusion that Sorbs2 is involved in fate specification of SHF progenitors is not supported.For example, RNA-seq data of differentiated hES cells suggest that knock-down of Sorbs2 reduces the expression of anterior SHF markers, such as Tbx1, and increases the expression of posterior SHF markers such as Tbx5, thus, the imbalance could be between aSHF and pSHF, rather than between FHF and SHF as suggested by the authors. Thus, is TBX5 expression altered in the pSHF of mutant embryos? How is the expression of other pSHF markers? Where is Sorbs2 expressed?

As per reviewer’s suggestion, we have done analysis on *^-/-^Sorbs2^-/-^* embryos. First, we checked DMP formation in E10.5 *^-/-^Sorbs2^-/-^* embryos. Consistent with E18.5 phenotype, we observed two types of DMP malformations.

Excerpt from result section:

“We also noted DMP malformation in 5 out of 15 E10.5 *Sorbs2^-/-^* embryos. Themajority of them had DMP aplasia/hypoplasia (n=4), whereas one of them had duplicated DMP (Figure 2D). The dichotomy of DMP morphology is consistent with two opposite ASD phenotypes seen in E18.5 embryos.”

As per reviewer’s suggestion, we did RNA-seq with *Sorbs2^-/-^* embryos. Data have shown decreased pSHF markers (Figure 3F and 3G).

Excerpt from result section:

“The decrease of posterior SHF marker genes were less general but still clearlydown-regulated in the majority of *Sorbs2^-/-^* embryos (Figure 3F). Decreased Tbx5 expression in posterior SHF of *Sorbs2^-/-^* embryos was validated by RNA in situ hybridization (n=3, Figure 3G).”

As per reviewer’s suggestion, we examined Sorbs2 expression with published datasets.

Excerpt from result section:

“To examine Sorbs2 expression pattern in early mouse embryos, we pooled publiclyavailable single-cell transcriptomic profiles from E9.25 to 10.5 mouse embryonic hearts(17, 18). We identified 9 subgroups as cardiac progenitors or myocardium and noted that Sorbs2 is highly expressed in cardiomyocytes and in a subgroup of cardiac progenitors that also express Isl1 and Tbx1 (Figure S5).”

It is virtually impossible to predict mechanisms leading to cardiac structural defects using hES differentiation data. Even though the mouse phenotype is incompletely penetrant, it should still be possible to analyse it in more details and validate the proposed c-ABL/NOTCH/SHH pathway in vivo.

As per reviewer’s suggestion, we validated in vitro findings in mouse embryos. Mouse embryo data are consistent with in vitro results.

Excerpt from result section:

“We used qPCR to validate molecular findings of hESCs in E10.5 mouse embryos.Since the penetrance of cardiac phenotype is about 40%, we increased the number of *Sorbs2^-/-^* embryos according to this ratio. Consistent with hESC differentiation results, we detected significantly down-regulated expression in 3 out of 4 Notch and Shh signaling target genes (Hey1, Heyl and Ptch1) (Figure 3A). Multivariate PERMANOVA (permutational multivariate analysis of variance) analysis revealed a significant combined difference in Notch and Shh signaling target expression between wild type and *Sorbs2^-/-^* embryos (p<0.009).”

Excerpt from result section:

“A portion of embryos had obvious down-regulation of Notch and Shh signalingtargets (Figure 3H), which is consistent with a low penetrance of ASD in homozygous mutants. RNA in situ hybridization confirmed decreased Heyl expression in posterior SHF of *Sorbs2^-/-^* embryos (n=1, Figure 3I). Next, we verified the upstream molecular changes of Notch signaling found in hESCs differentiation model. Indeed, we noted that Notch1 expression level significantly decreased whereas c-ABL significantly increased in *Sorbs2^-/-^* embryos (Figure 3I-3J).”

– Human genetics studies were performed on a relatively small number of patients; nevertheless they provided some suggestion that SORBS2 may have a role in CHD. The authors should show a map of the gene model or predicted protein with the location of the variants and list the actual variants that they found.Also, did the 20 patients that carried "rare damaging variants" of SORBS2 also have variants in other CHD genes?

We have listed the location of variants in gene and protein in Figure 4C.

The majority of patients with rare *SORBS2* variants also have other CHD gene variants. They are listed in Table S6.

Reviewer #2:– My main concern is related to the arguments used to claim a role of SORBS2 in SHF development through c-ABL/NOTCH/SHH axis.

As per reviewer’s suggestion, we have validated our in vitro findings in mouse embryos.

Excerpt from result section:

“We used qPCR to validate molecular findings of hESCs in E10.5 mouse embryos.Since the penetrance of cardiac phenotype is about 40%, we increased the number of *Sorbs2^-/-^* embryos according to this ratio. Consistent with hESC differentiation results, we detected significantly down-regulated expression in 3 out of 4 Notch and Shh signaling target genes (Hey1, Heyl and Ptch1) (Figure 3A). Multivariate PERMANOVA (permutational multivariate analysis of variance) analysis revealed a significant combined difference in Notch and Shh signaling target expression between wild type and *Sorbs2^-/-^* embryos (p<0.009).

To have a view of transcriptomic changes in Sorbs2 mutants, we performed RNA-seq on wild type, *Sorbs2^+/-^* and *Sorbs2^-/-^* embryos. PCA analysis indicates that wild type embryos are clustered together, whereas *Sorbs2^-/-^* embryos are more scattered (Figure 3B), which is consistent with diverged cardiac phenotypes of *Sorbs2^-/-^* embryos. Interestingly, the majority of Sorbs2+/- samples are juxtaposed more closely to *Sorbs2^-/-^* embryos in PCA plot, indicating a molecular phenotype in heterozygous mutants. We selected genes significantly down-regulated in heterozygous mutants (log2(fold change)>0.25, p<0.05) to perform GO analysis and found that these genes are enriched in pathways involved in muscle development and embryonic morphogenesis (Figure 3C, Table S4). Using the same threshold, we selected genes significantly down-regulated in homozygous mutants to perform GO analysis. These genes are enriched in pathways regulating heart development, myofibril assembly and cardiac septum development (Figure 3D, Table S5). Particularly, the Notch signaling pathway was also in the enrichment list. Looking closer, we noted that genes involved in cardiac development, myofibril assembly and contraction force were down-regulated in nearly all the homozygous mutants (Figure 3E). The decrease of posterior SHF marker genes were less general but still clearly down-regulated in the majority of *Sorbs2^-/-^* embryos (Figure 3F). Decreased Tbx5 expression in posterior SHF of *Sorbs2^-/-^* embryos was validated by RNA in situ hybridization (n=3, Figure 3G). A portion of embryos had obvious down-regulation of Notch and Shh signaling targets (Figure 3H), which is consistent with a low penetrance of ASD in homozygous mutants. RNA in situ hybridization confirmed decreased Heyl expression in posterior SHF of *Sorbs2^-/-^* embryos (n=1, Figure 3I). Next, we verified the upstream molecular changes of Notch signaling found in hESCs differentiation model. Indeed, we noted that Notch1 expression level significantly decreased whereas c-ABL significantly increased in *Sorbs2^-/-^* embryos (Figure 3J-3K).”

– It is clear that Sorbs2-null mice have ASD. However, the authors did not find the origin of this defect. The authors should show how dorsal mesenchymal protrusion is formed in these mutant embryos.

As per reviewer’s suggestion, we have done morphological analysis on E10.5 embryos. Results indicate that, indeed, there are two types of DMP malformation in E10.5 *Sorbs2^-/-^* embryos (Figure 2D), which is consistent with E18.5 phenotypes.

Excerpt from result section:

“We also noted DMP malformation in 5 out of 15 E10.5 *Sorbs2^-/-^* embryos. Themajority of them had DMP aplasia/hypoplasia (n=4), whereas one of them had duplicated DMP (Figure 2D). The dichotomy of DMP morphology is consistent with two opposite ASD phenotypes seen in E18.5 embryos.”

– The use of shRNA has some limitations that the authors did not discuss. Reduction of SORBS2 expression resulted in reduction of cTnT-positive cells. However, the authors should show whether apoptosis is normal in shRNA treated cells.

As per reviewer’s suggestion, we have done TUNEL staining on our cell lines. We didn’t detect difference in apoptosis between control and shRNA knockdown ES cells (Figure S1F).

Excerpt from result section:

“SORBS2-knockdown did not affect clone morphology, pluripotency markerexpression and apoptosis of hESCs (Figure S1B-S1F).”

– The expression of sarcomeric genes is reduced in SORBS2-KD cells. How do the authors explain this result?

Other than the reduced differentiation efficiency in SORBS2 KD cells, the impaired maturation of differentiated cardiomyocytes due to abnormal sarcomere assembly may also contribute to reduced sarcomeric gene expression. We observed the same outcome in Sorbs2 mouse mutants.

– In this study the qPCR is used to examine transcriptional levels of FHF and SHF genes during cardiomyocyte differentiation. Increase of TBX5, HAND1 and HCN4 is observed at D5 but clearly significant at D10. I'm wondering whether these markers signed the FHF or the atrial identity? The authors should validate this observation using other FHF markers.

As per reviewer’s suggestion, we looked the expression of other FHF markers, such as *GATA4* and *TBX2*, in RNA-seq data. Both genes were significantly up-regulated in D5 *SORBS2*-knockdown cells.

**Author response table 1. resptable1:** 

Gene name	Base Mean	Log2FoldChange	LfcSE	Stat	pvalue	padj
GATA4	7197.309	0.330043	0.078422	4.208528	2.57E-05	0.000904
TBX2	1295.333	0.58354	0.122726	4.754828	1.99E-06	0.000111

The current work is to establish the role of *Sorbs2* in promoting SHF development rather than a role in maintaining the balance between FHF and SHF. We have removed this hypothetical statement, ”…suggesting that *SORBS2* is a critical regulator to maintain the balance between FHF and SHF cells during differentiation”. As a short report, we have used all the space for this major aim. We respectfully request reviewers and editors not to ask these data.

– qPCR shows a decrease of three SHF genes. However, there is no SHF markers (except FGF8) listed in TableS2 (downregulated genes at D5 from RNA-seq analysis). The authors should discuss this point.

DEGs in Table S2 are genes used for GO analysis. Since we had a large number of DEGs between D5 control and *SORBS2*-knockdown cells. We used very stringent criteria to select genes (padj. < 0.05, |log2(fold change)| > 1) for GO analysis. Actually, other SHF markers were also significantly down-regulated in RNA-seq data. Since they are below the threshold, they were not included in Table S2. The Author response table 2 lists the data on other SHF markers.

**Author response table 2. resptable2:** 

Gene name	Base Mean	Log2FoldChange	LfcSE	Stat	pvalue	padj
TBX1	721.4109	-0.99858	0.116671	-8.55897	1.14E-17	1.15E-14
ISL1	2401.652	-0.38996	0.091506	-4.26157	2.03E-05	0.000749
CXCR4	145.2801	-0.91179	0.213437	-4.27194	1.94E-05	0.000723
MEF2C	1242.792	-0.22804	0.107697	-2.11741	0.034225	0.216401

– I do not understand why the authors focused on the c-ABL/NOTCH/SHH axis to explain the role of SORBS2 in SHF development. The arguments that link NOTCH to SHH signaling pathways in this context are very weak. The mechanism is incomplete and here again the mouse model should have been used to validate these observations.

Notch signaling pathway is the only signal pathway that shows up in both GO analyses with cardiomyocyte differentiation and mouse embryo RNA-seq data. Existing literature indicates that Sorbs2 can regulate Notch protein level in cell and fly models (refs. 11-13). Our work validated that similar regulation exists in both cardiomyocyte differentiation and mouse embryo models.

Notching signaling is essential for SHF development (Chien-Jung L. etc. *Development*; 139: 3277-3299). Alagille syndrome patients have both outflow and inflow defects. Mouse mutants with genetic defects in Notch signaling components recapitulate cardiac defects of Alagille syndrome patients. In addition, we detected decreased expression of Shh signaling targets. Shh signaling is also essential for DMP formation (Goddeeris MM etc. Development; 135:1887-1895, Hoffmann AD etc. Development; 136: 1761-1770). Since Notch signaling is a known molecular mechanism promoting Smo accumulation in cilia and enhancing cellular response to Shh (ref.14), we proposed the current working model. We believe that both Notch and Shh signaling defects contribute to ASD pathogenesis. More important, we validated the molecular mechanism in mouse embryos (Figure 3, Table S4 and S5).

Certainly, there might have other molecular changes that contribute to ASD pathogenesis. For example, the decreased *Tbx1* expression may contribute to ASD as well. *Tbx1* deficiency also leads to DMP malformation and ASD (Rana MS etc. *Circ.Res;* 115:790–799).

– The authors sequenced 300 CHD cases and found several missense variants in SORBS2 gene. TableS6 shows heart phenotype in patients. "The majority of patients with rare SORBS2 variants had ASD". The authors should show whether there is any correlation between SORBS2 mutation and ASD phenotype.

Thanks for the reviewer’s insight. We did detect a significant enrichment of SORBS2 variants in patients with ASD.

Excerpt from result section:

“In our CHD cohort, we noted a significant enrichment of SORBS2 rare damaging variants in patients with ASD (17 out of 183 ASD patients versus 3 out of 117 non-ASD patients, p=0.0306, Fisher's exact test).”

Reviewer #3:In this study, the Authors have attempted to use a mouse model to link the cardiac malformations observed in patients with the 4q deletion to defects in the SORBS2 locus. It is interesting to see that 40% of ^-/-^Sorbs2^-/-^ hearts had ASD, and that some of them have an absence/hypoplasia of the primary septum and/or double atrial septum with an incomplete penetrance. However, while the ratio of cardiac defects in embryos is similar to the percentage of reported postnatal lethality, in general ASD hardly causes 100% postnatal lethality and likely cannot provide an adequate explanation as the sole cause of the early postnatal death. In addition, ASDs were only seen in the case of null genotypes (Sorbs2^-/-^), and the authors didn't report ASDs in the case of Sorbs2^+/-^ genotypes. This is crucial because in 4q-, SORBS2 is going to be haploinsufficient, and not a null genotype.

We have changed the lethality statement from “…may cause the early postnatal death” to “…might contribute to the early postnatal death”.

Excerpt from result section:

“The penetrance of ASD is similar to the ratio of reported postnatal lethality,indicating that ASD might contribute to the early postnatal death.”

The reason we didn’t detect ASD in heterozygous mutant might be due to different dosage sensitivity between different species. Indeed, we found that the gene expression profile of heterozygous mutant is closer to that of homozygous mutants than to wild type. It indicates similar molecular defects in heterozygous mutants (Figure 3B-3C, Table S4). Besides, human population has very heterogenous genetic background. other 4q interval gene haploinsufficiency and genetic modifiers in other loci may work together with SORBS2 haploinsufficiency to cause defects in human population, in other words, increase the penetrance of SORBS2 haploinsufficiency.

The major fault of the study is, if the authors feel that SORBS2^+/-^ haploinsufficiency is the cause for cardiac defects, patient samples (from 4q- or SORBS2 VUCS) should be used to derive iPSCs instead of the hRNA knockdown ESCs for all subsequent studies.

This work is to dissect the role of *SORBS2* in CHD pathogenesis of 4q deletion syndrome. Since *SORBS2* is entirely deleted in 4q deletion. It would be nicer if we could have patients with heterozygous *SORBS2* null allele. Unfortunately, we haven’t identified such patient. We respectfully request reviewers and editors not to ask this data. Instead, we used shRNA knockdown cell lines that express *SORBS2* at ~40% of wild type level.

SORBS2-knockdown iPSCs were used for cardiomyocyte differentiation, which cannot be directly linked to cardiac defects such as ASD. Since Sorbs2^-/-^ is available, the molecular mechanisms should be tested with an in vivo model or primary cell cultures derived from Sorbs2^-/-^ or Sorbs2^+/-^ models.

As per reviewer’s suggestion, we have validated our in vitro findings with mouse embryos.

Excerpt from result section:

“We used qPCR to validate molecular findings of hESCs in E10.5 mouse embryos.Since the penetrance of cardiac phenotype is about 40%, we increased the number of *Sorbs2^-/-^* embryos according to this ratio. Consistent with hESC differentiation results, we detected significantly down-regulated expression in 3 out of 4 Notch and Shh signaling target genes (Hey1, Heyl and Ptch1) (Figure 3A). Multivariate PERMANOVA (permutational multivariate analysis of variance) analysis revealed a significant combined difference in Notch and Shh signaling target expression between wild type and *Sorbs2^-/-^* embryos (p<0.009).

To have a view of transcriptomic changes in Sorbs2 mutants, we performed RNA-seq on wild type, Sorbs2+/- and *Sorbs2^-/-^* embryos. PCA analysis indicates that wild type embryos are clustered together, whereas *Sorbs2^-/-^* embryos are more scattered (Figure 3B), which is consistent with diverged cardiac phenotypes of *Sorbs2^-/-^* embryos.

Interestingly, the majority of Sorbs2+/- samples are juxtaposed more closely to *Sorbs2^-/-^* embryos in PCA plot, indicating a molecular phenotype in heterozygous mutants. We selected genes significantly down-regulated in heterozygous mutants (log2(fold change)>0.25, p<0.05) to perform GO analysis and found that these genes are enriched in pathways involved in muscle development and embryonic morphogenesis (Figure 3C, Table S4). Using the same threshold, we selected genes significantly down-regulated in homozygous mutants to perform GO analysis. These genes are enriched in pathways regulating heart development, myofibril assembly and cardiac septum development (Figure 3D, Table S5). Particularly, the Notch signaling pathway was also in the enrichment list. Looking closer, we noted that genes involved in cardiac development, myofibril assembly and contraction force were down-regulated in nearly all the homozygous mutants (Figure 3E). The decrease of posterior SHF marker genes were less general but still clearly down-regulated in the ^major^ity of *Sorbs2^-/-^* embryos (Figure 3F). Decreased Tbx5 expression in posterior SHF of *Sorbs2^-/-^* embryos was validated by RNA in situ hybridization (n=3, Figure 3G). A portion of embryos had obvious down-regulation of Notch and Shh signaling targets (Figure 3H), which is consistent with a low penetrance of ASD in homozygous mutants. RNA in situ hybridization confirmed decreased Heyl expression in posterior SHF of *Sorbs2^-/-^* embryos (n=1, Figure 3I). Next, we verified the upstream molecular changes of Notch signaling found in hESCs differentiation model. Indeed, we noted that Notch1 expression level significantly decreased whereas c-ABL significantly increased in *Sorbs2^-/-^* embryos (Figure 3J-3K).”

[Editors’ note: what follows is the authors’ response to the second round of review.]

Essential Revisions:There are two sets of experiments that are deemed critical (please see details in the individual reviews):1. A rescue experiment using recombinant NOTCH or SHH in the cell culture model.2. Test additional pSHF markers by in situ hybridization on the DMP or pSHF of mutant embryos to validate and regionalize at least some of the expression changes revealed by whole-embryo RNA-seq.

We thank the editors and reviewers for taking the time to evaluate our manuscript. As per suggestion, we have performed rescued experiment with recombinant Shh protein in in vitro cardiomyocyte differentiation model and performed in situ hybridization with an additional pSHF marker. Please check details in the following responses to reviewers.

eLife's editorial process also produces an assessment by peers designed to be posted alongside a preprint for the benefit of readers.Reviewer #1:The resubmitted version has addressed the issues raised by the previous round of review, albeit some questions remain.I have 3 recommendations for authors:1. Phenotypic analysis in the mouse mutant has been extended, but it is still limited. However, the authors did find developmental abnormalities with the DMP, which is interesting (albeit predictable). Phenotyping would benefit from testing additional pSHF markers and perhaps by testing basic mechanisms such as cell proliferation and cell death

As per reviewer’s suggestion, we have tested an additional pSHF marker *Hoxb1* and found that it was downregulated in pSHF (3 out of 4 embryos).

Excerpt from Results

“Decreased Tbx5 (n=3) and Hoxb1 (n=3 out of 4) expression in posterior SHF of *Sorbs2^-/-^* embryos was validated by RNA in situ hybridization (Figure 3G and 3H).”

2. Mechanisms: the focus of the authors in a so called c-Abl-Notch1-Shh "axis" remains speculative and based on gene (and in some cases protein) expression assays in differentiated hES cells and in whole embryo RNA-seq. These results are insufficient to draw conclusions about the pSHF or whatever it is happening in the DMP. Claims to the effect that Sorbs2 specifies "the fate of second heart field (SHF) progenitors through c-ABL/NOTCH/SHH axis" are unsubstantiated and should be removed (in the abstract, subheading of results, discussion), or be proposed as a speculative hypothesis in the discussion. In addition, the authors show no evidence that Sorbs2 specifies the fate of SHF progenitors. On the other hand, new data added to the resubmission clearly show a downregulation of Tbx5 in the DMP/pSHF region of mutants. It is unclear whether this is due to hypoplasia of the DMP or to reduced expression of the gene. In any case, this could be a valuable clue to understand mechanisms as Tbx5 is a critical factor in DMP development (e.g. Xie et al., 2012, Moskowitz group, which has also shown interaction between Tbx5 and SHH signalling), thus it could be part of the pathogenesis of ASD in this mutant. This should be discussed and contrasted with the NOTCH hypothesis.

We agree with Reviewer 1 that our work lacks of embryonic evidence to substantiate our previous claim. As per reviewer’s suggestion, we have removed the statement, “specifying the fate of SHF through c-ABL/NOTCH/SHH axis”, from the Abstract and subheading of Results. We state it as a hypothesis in discussion.

We thank the reviewer for drawing our attention to the important *Tbx5* works and point out additional molecular explanation for the pathogenesis of *Sorbs2* mutants. Now we have cited this article and discussed the possible role of *Tbx5* in the pathogenesis of *Sorbs2* mutants.

Excerpt from Discussion

“In both models, we detected increased protein level of NOTCH1 endocytosis facilitator c-ABL, decreased NOTCH1 protein level and impaired SHH signaling. Notch and Shh signaling are essential for SHF development(24). Notch signaling promotes Smo accumulation in cilia and enhancing cellular response to Shh, placing Notch upstream of Shh signaling (16, 25). Therefore, SORBS2 might promote SHF progenitor fate through c-ABL/NOTCH/SHH axis. In addition, Tbx5 was also significantly downregulated in posterior SHF of Sorb2 mutants. Tbx5-Hh molecular network is an essential regulatory mechanism in SHF for atrial septation(26). It is likely that Tbx5 downregulation also contributes to the pathogenesis of ASD through its effect on Hh signaling. Consistently, adding recombinant SHH protein is sufficient to rescue SHF marker gene expression and cardiomyocyte differentiation efficiency.”

3. The authors should discuss the apparently inconsistent findings of increased expression of Tbx5 in the hES cell model, and the decreased expression of the same gene in the pSHF of mutant embryos.

As per reviewer’s suggestion, we have added the relevant discussion about the discrepancy between in vitro and in vivo model. Based on literature, Tbx5 mainly functions as a FHF regulator in in vitro cardiogenesis (Anderson etc. Nat. Commun. 2018; 9: 3140). The simplified in vitro model may not fully recapitulate the multiple roles of Tbx5 in regulating cardiogenesis in vivo.

Excerpt from Discussion

“However, TBX5 was upregulated in D5 SORBS2 knockdown cells. It is likely that in vitro cardiomyocyte differentiation is a simplified model that can not fully recapitulate the in vivo spatial and temporal information of various cardiac progenitors and therefore has less regulatory layers in which TBX5 may predominantly function as a FHF regulator. Indeed, Tbx5 knockdown reduces FHF progenitors and has no effect on SHF progenitors in an in vitro cardiogenesis model(27).”

Reviewer #3:The authors have replied to several of my concerns such as the arguments that SORBS2 specifies SHF progenitors through c-ABL/NOTCH/SHH. They used PCR, RNA-seq and in situ hybridization to validate their hypothesis. Thus, they showed that Notch1 protein expression level is significantly decreased whereas c-ABL is increased in ^-/-^Sorbs2^-/-^ mutant embryos. In this revised version, they better used their in vivo model to support their hypothesis. The phenotype of the DMP is now examined which is consistent with ASD observed at E18.5. Control quality of shRNA KD hES cells is now performed and no apoptosis is detected. Expression of pSHF markers is now shown including Hoxb1, Hoxa1, Aldh1a2 and Tbx5 which confirms the previous statement. The rational to examine NOTCH and SHH signals is now better explained. Finally, human genetics data is improved and detection of significant enrichment of SORBS2 rare damaging variant in patients with ASD vs non-ASD is great. However, I agree with reviewer#3 that hiPS cells model rather than hESC would have been better to support their hypothesis. In addition, a rescue experiment using recombinant NOTCH or SHH treatment in the in vitro culture would have been important to perform in order to support the main finding of this study.

We thank the reviewer’s constructive suggestion that would strengthen our conclusion. We have performed rescue experiment with recombinant SHH protein. Results showed that it was sufficient to upregulate SHF markers gene expression and improve the efficiency of cardiomyocyte differentiation, suggesting that SORBS2’s function is mediated through SHH signaling.

Excerpt from Results

“We applied recombinant SHH protein to check whether it can rescue defects caused by SORBS2 knockdown. As expected, exogenous SHH activated PTCH1 and GLI1 expression (Figure 1O). It also upregulated the expression of SHF marker ISL1 and MEF2C in D5 SORBS2-shRNA1 cells (Figure 1O) and rescued cardiomyocyte differentiation efficiency with more cells presenting a polygonal or spindle-like shape (Figure 1P and 1Q).”